# Metal chalcogenide hollow polar bipyramid prisms as efficient sulfur hosts for Na-S batteries

Muhammad Kashif Aslam [1], Ieuan D. Seymour[2], Naman Katyal[2], Sha Li[3], Tingting Yang[1], Shu-juan Bao[1], Graeme Henkelman [2✉] & Maowen Xu [1✉]

Sodium sulfur batteries require efficient sulfur hosts that can capture soluble polysulfides and enable fast reduction kinetics. Herein, we design hollow, polar and catalytic bipyramid prisms of cobalt sulfide as efficient sulfur host for sodium sulfur batteries. Cobalt sulfide has interwoven surfaces with wide internal spaces that can accommodate sodium polysulfides and withstand volumetric expansion. Furthermore, results from in/ex-situ characterization techniques and density functional theory calculations support the significance of the polar and catalytic properties of cobalt sulfide as hosts for soluble sodium polysulfides that reduce the shuttle effect and display excellent electrochemical performance. The polar catalytic bipyramid prisms sulfur@cobalt sulfide composite exhibits a high capacity of 755 mAh g$^{-1}$ in the second discharge and 675 mAh g$^{-1}$ after 800 charge/discharge cycles, with an ultralow capacity decay rate of 0.0126 % at a high current density of 0.5 C. Additionally, at a high mass loading of 9.1 mg cm$^{-2}$, sulfur@cobalt sulfide shows high capacity of 545 mAh g$^{-1}$ at a current density of 0.5 C. This study demonstrates a hollow, polar, and catalytic sulfur host with a unique structure that can capture sodium polysulfides and speed up the reduction reaction of long chain sodium polysulfides to solid small chain polysulfides, which results in excellent electrochemical performance for sodium-sulfur batteries.

[1] Key Laboratory of Luminescence Analysis and Molecular Sensing (Southwest University), Ministry of Education, School of Materials and Energy, Southwest University, Chongqing 400715, PR China. [2] Department of Chemistry, The University of Texas at Austin, Austin, TX 78712, USA. [3] State Key Laboratory of Physical Chemistry of Solid Surface, Collaborative Innovation Center of Chemistry for Energy Materials, Department of Chemistry, College of Chemistry and Chemical Engineering, Xiamen University, Xiamen 361005, China. ✉email: henkelman@utexas.edu; xumaowen@swu.edu.cn

Research in energy storage and energy conversion systems is important to develop technologies that can contribute to resource sustainability and long-term development of human society by studying new electrode materials with high energy and power densities. Group IA alkali–metal sulfur (Li–S and Na–S) batteries have been investigated extensively due to their high energy density and energy storage properties[1–7]. Na–S batteries have charge/discharge mechanisms similar to Li–S batteries that involve a two-electron redox per sulfur atom, which is why its theoretical sulfur capacity will also be 1672 mAh g$^{-16-8}$. Similar to Li–S, Na–S batteries offer high energy density, power density and energy efficiency, and are nontoxic with good cycle life. An advantage of Na–S batteries over Li–S batteries is the low cost and high abundance of the raw materials that make them suitable for large-scale production[8,9]. In the last decade, two operating temperature ranges of Na–S batteries have been studied: high-temperature 270–350 °C and room temperature. At high operating temperatures, the molten sodium metal and sodium polysulfides (NaPSs) are known to be corrosive and can cause vigorous reactions between the molten sodium and molten sulfur through a broken separator film[10]. Safety concerns are very important for energy storage technologies, as exemplified by the recent fire incident at NGK Insulator Ltd. related to a Na–S battery[8]. In contrast, at ambient temperatures, Na–S batteries are safer, more durable and offer lower operating costs and thus have been studied by many research groups[9–12]. Nevertheless, Na–S batteries face problems, including: (1) the dissolution of NaPSs in the electrolyte causing high capacity fade; (2) formation of sodium metal dendrites during charging that inevitably lead to short-circuits; and (3) the low utilization rate of the sulfur cathode, that constrains the electrochemical reactions to a certain distance from the current collector and limits the cell capacity.

Specifically, at room temperature, Na–S batteries have some crucial problems that need to be addressed. Firstly, the poor conductivity of the active material, S, results in slow electrochemical reaction kinetics and lower S utilization[13,14]. Secondly, Na–S batteries have a higher volume strain rate than Li–S batteries, which makes the cathode of Na–S batteries susceptible to collapse[15]. Finally, NaPSs produced during the multi-step discharge reaction have high solubility and reactivity, which makes it easier for NaPSs to diffuse toward the Na–anode and cause a severe "shuttle effect", resulting in a significant decrease in capacity[5]. Nevertheless, some progress has been made to improve the conductivity of S cathodes and speed up the reaction kinetics of Na–S batteries at room temperature. For example, carbon materials can be combined with S to improve the cathodic electrical conductivity, including carbon spheres[16], carbon nanofibers[17], carbon nanotubes[18], carbon cloths[19], etc. These materials not only improve the utilization rate of S, but also control the volume strain due to their rich porous structure. Zhang et al. assembled a Na–S battery using S-interconnect hollow nanospheres of carbon (S@Con-HC) as a cathode, but it displays limited cycling stability[5]. Zhang et al. constructed a Na–S battery using porous carbon microspheres (S-PCMs) as the cathode but it also suffers from inferior redox activity[18]. Although the S/C hybrid designs can greatly improve the utilization of sulfur, it should be pointed out that it is very difficult to achieve full electrochemical reversibility using nonpolar carbon structures as S hosts. Nonpolar carbon-based materials have weak interactions with polar NaPSs and cannot effectively prevent the diffusion of NaPSs, resulting in NaPSs ($Na_2S_x$, $4 \le x \le 8$) moving between the anode and cathode (the shuttle effect), which results in parasitic redox reactions on the surface of the Na–metal anode (Fig. 1b).

In recent years, polar materials, including $MnO_2$[20], $TiO_2$[21], alpha-$Ni(OH)_2$[22], and $SiO_2$[23] have been shown to strongly interact with polysulfides and effectively bind them to the cathode

to attain reversible electrochemical cycling. Nevertheless, most of the metal oxides and hydroxides are semiconductors with poor electronic conductivities, resulting in low rate capabilities. Suitable sulfur hosts should have both strong polysulfide attraction and good electronic conductivity[24]. Moreover, since the interaction between solid polar (non-hollow) materials and NaPSs is created only on the surface as single-layer chemical adsorption, such solid structures can only capture a small amount of NaPSs; NaPSs away from the surface can still dissolve in the electrolyte and shuttle between the electrodes (Fig. 1c)[25]. A promising approach is to construct a polar hollow architecture, which can physically block the path of outward diffusion, thus encapsulating the polysulfides in the wide inner space of the cathode (Fig. 1d). As a result, hollow polar S hosts can more efficiently block NaPSs diffusion than other structures, such as nanoparticles and flakes. In addition to the above strategies, the use of host materials that effectively catalyse the conversion of long-chain NaPSs ($Na_2S_x$, $4 \le x \le 8$) to short chain NaPS is a particularly promising approach to inhibit NaPSs diffusion[26]. Due to the insulating properties of sulfur and NaPSs, the electrochemical discharge/charge process is sluggish. For non-catalytic hosts such as carbon, carbon nanotubes, carbon nano fibers, carbon hollow nanospheres, and double-shell carbon microspheres, the conversion of NaPSs is slow and the intermediate polysulfides can easily dissolve into the electrolyte. However, due to the use of the catalytic S host, such as electronically conducting transition metal sulphide hosts can effectively act as electrocatalysts to accelerate the redox kinetic of long chain NaPSs ($Na_2S_x$, $4 \le x \le 8$) and efficiently convert to solid phase S and $Na_2S/Na_2S_2$ (Fig. 1d). Consequently, the use of catalytic materials as sulfur carriers in the construction of advanced Na–S batteries could be a promising approach.

Here, inspired by the above encouraging strategies of hollow, polar and catalytic S hosts, we synthesized a unique architecture, hollow polar bipyramid prism catalytic $CoS_2$/C as a sulfur carrier (S@BPCS) (Fig. 1a). S@BPCS is an efficient host for solid-phase S and NaPSs, because every bipyramid prism structure has a wide hollow cavity for sulfur loading and for the entrapment of NaPSs. When used as a cathode in Na–S batteries, S@BPCS exhibits good electrochemical performance with stable cycling and rate capability that suggests maximum utilization of S, strong binding of NaPSs, is catalytically active for converting long-chain polysulfides into small chain polysulfides, and it delivers good electrical conductivity. Here, the charge/discharge mechanisms of S@BPCS for reversible battery reactions have been studied using in situ Raman spectroscopy, in situ XRD, ex situ TEM, ex situ X-ray photoelectron spectroscopy (XPS), and density functional theory (DFT) calculations.

## Results

**Synthesis and characterization of sulfur hosts.** Inspired by the advantages of hollow, polar, interweaving surface structure, and catalytic activity of Co for Na–S batteries, we synthesized hollow polar bipyramid prism cobalt chalcogenides and used them as sulfur hosts. The bipyramid prisms of cobalt were synthesized by simple reflux methods (see "Methods" section). The X-ray diffraction (XRD) pattern confirmed the formation of $C_{10}H_{16}Co_3O_{11}$ and was attributed to the standard PDF of JCPDS#22-0582, as shown in Supplementary Fig. 1.

In order to study the structure and morphology of the as-prepared precursor, field emission scanning electron microscopy (FESEM) (Fig. 2a, b) and transmission electron microscopy (TEM) (Fig. 2c and Supplementary Fig. 2) along with energy dispersive spectroscopy (EDS) elemental mapping was performed (Fig. 2d). The bipyramid Co prisms were converted into hollow bipyramid Co/C prisms. To confirm the phase purity and hollow

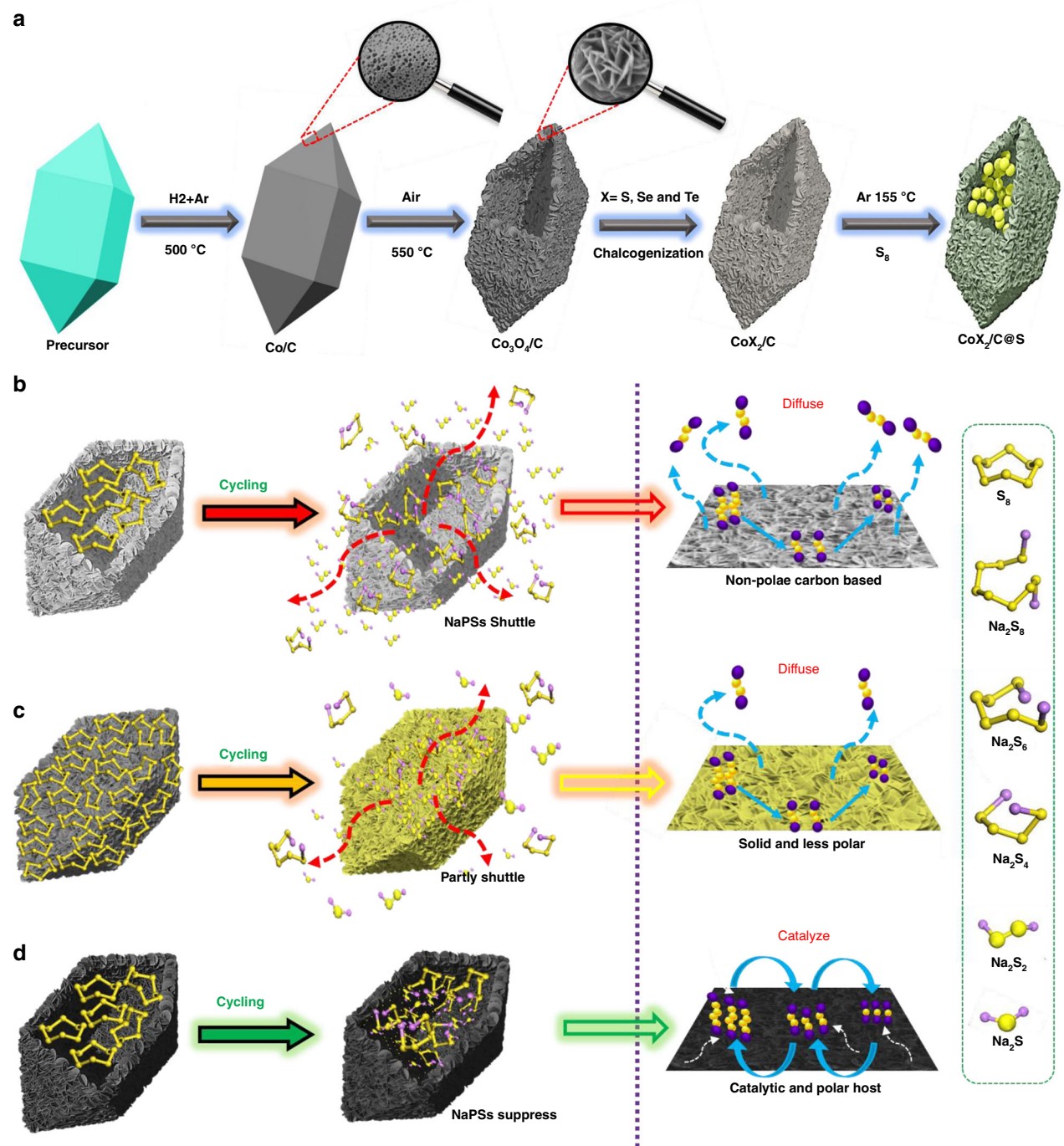

**Fig. 1 Synthesis of chalcogenides and illustration of polysulfide diffusion effect. a** Synthesis of metal chalcogenide S@BPCX composites. **b** Nonpolar hollow carbon host. **c** Diffusion of NaPSs in solid nonpolar host and **d** suppression of NaPSs in hollow polar/catalytic host.

interior of Co/C, XRD and TEM tests were performed (Supplementary Fig. 3 and Supplementary Fig. 4b), respectively. Afterward, the Co/C hollow prisms were converted into $Co_3O_4/C$ (BPCO). The phase purity and morphology of BPCO were investigated by XRD and FESEM, as shown in Supplementary Fig. 5 and Fig. 2e, f, respectively. FESEM images of BPCO show that it has a unique hierarchical surface with interweaving plate-like structures. To confirm the elemental composition, EDS mapping was performed, as shown in Supplementary Fig. 6. Furthermore, BPCO was converted into $CoS_2/C$ (BPCS), $CoSe_2/C$ (BPCSE) and $CoTe_2/C$ (BPCTE) by an ion-exchange method, their formation and phase purity was confirmed by XRD analysis

as represented in Supplementary Figs. 7–9 and their chemical compositions was confirmed by EDS elemental mapping as shown in Supplementary Figs. 10–12, respectively. Figure 2g shows a FESEM image of BPCS representing a distinctive morphology with a hierarchical surface.

Furthermore, the chemical composition and morphology of BPCS was confirmed by TEM; Fig. 2h shows that the bipyramid hollow prisms of BPCS have 376 nm hollow wide space to accommodate sulfur and polysulfides within it. In addition, HRTEM and single-crystal spot transmission electron diffraction patterns confirm the synthesis of BPCS, as shown in Fig. 2i–k. The Brunauer–Emmett–Teller (BET) surface area of BPCS is

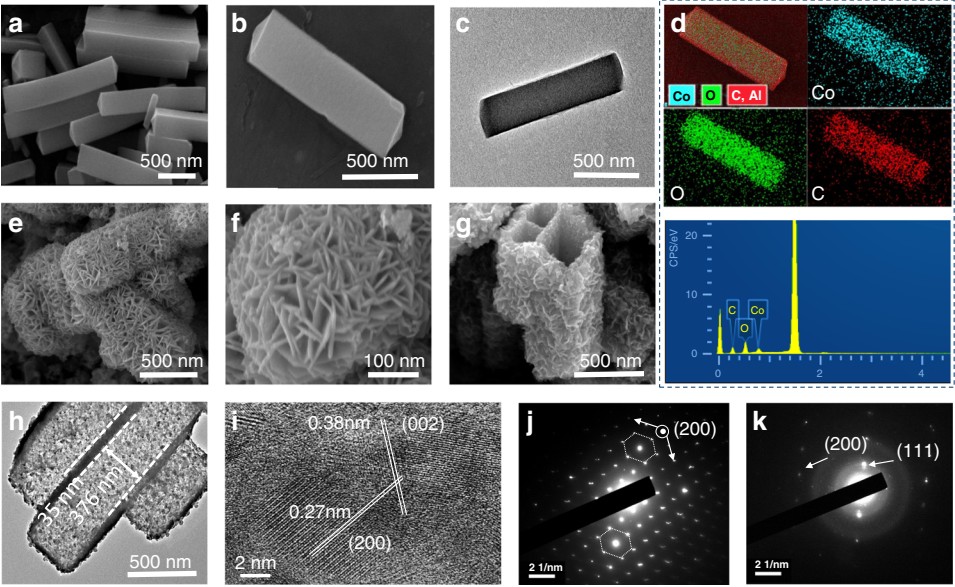

**Fig. 2 Illustration and characterization of bipyramid like prisms of co-precursor. a**, **b** FESEM images. **c** TEM image and **d** EDS elemental mapping of co-precursor. **e**, **f** FESEM images of BPCO. **g** FESEM image. **h** Low-resolution TEM image. **i** High-resolution TEM image and **j**, **k** SAED patterns of BPCS.

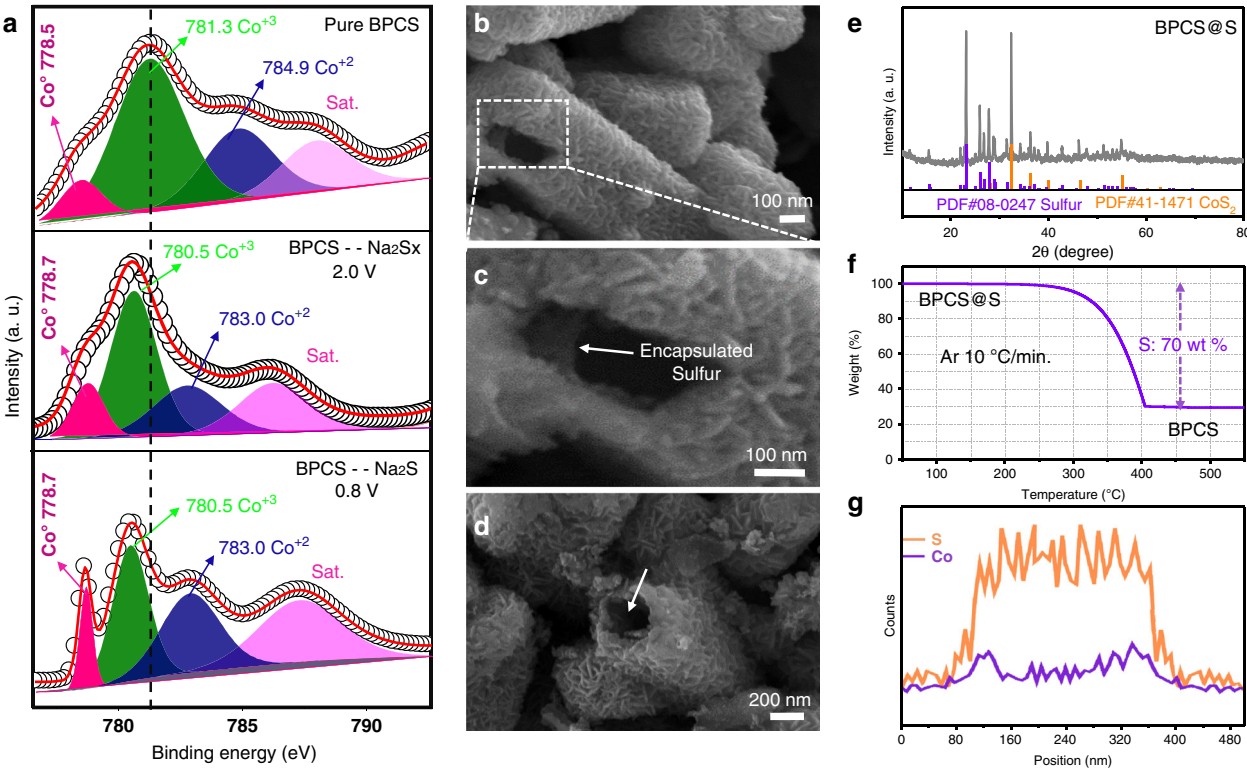

**Fig. 3 Characterization of bipyramid prism like S@BPCS composite. a** Comparison of the high resolution XPS of Co 2p3/2 of pure S@BPCS, after discharge at 2 V and 0.8 V. **b**–**d** FESEM images of S@BPCS composite. **e** XRD pattern. **f** TGA curve and **g** EDS line scan of S@BPCS.

$119.9 \, m^2 \, g^{-1}$ and the hysteresis loop is of type H4 indicates the physisorption isotherm of type I, which conforms to IUPAC, demonstrating that BPCS is composed of uniform slit-like pores, representing microporous structure, as shown in Supplementary Fig. 13. Finally, sulfur was injected into the hollow prisms using a melt-diffusion method to achieve the S@BPCS composite, as shown in Fig. 3b–d. Figure 3c shows that sulfur efficiently steamed into the inner side of the prism wall and there is no sulfur agglomeration on the hierarchical surface of the BPCS,

indicating sulfur removal from the surface at 190 °C. The XRD pattern of the S@BPCS composite confirms the cubic sulfur loading in the hollow bipyramid with interweaving plates like surfaced BPCS (Fig. 3e). An EDX line scan was recorded to examine the position of sulfur; Fig. 3g clearly shows sulfur efficaciously loaded inside the hollow prism. Furthermore, a thermogravimetric analysis (TGA) was performed to measure the content of loaded sulfur in the hollow prism S@BPCS composite. The mass of loaded sulfur was calculated using Fig. 3f

and Supplementary Fig. 14, and determined to be as high as 64.5%.

**Electrochemical performance of Na–S battery**. To investigate chemical interactions between BPCS and NaPSs, ex situ XPS analysis was employed, as shown in Fig. 3a. To understand the chemical interactions between the host BPCS and NaPSs we made three different samples for ex situ XPS as pure BPCS and two discharged cell samples at 2.0 and 0.8 V. The binding energies of cobalt decreases as electrons transfer from NaPSs to BPCS when discharged at 2.0 and 0.8 V, which is attributed to the strong chemical interaction between BPCS and NaPSs during the discharge process. This type of interaction between BPCS and NaPSs suppresses the shuttle effect, increases the cycling stability, and improves the capacity.

To examine the electrochemical storage properties and NaPSs redox reactions of polar catalytic BPCS and other metal chalcogenides, cyclic voltammetry (CV) and galvanostatic charge/discharge (GCD) plots were recorded as shown in Supplementary Fig. 15. Impressively, the CV plot of S@BPCS exhibits comparatively higher current density and peak potential because of its high electrical conductivity as well as catalytic and high polar nature that results in enhanced electrical conductivity and strong chemical interaction between BPCS and NaPSs. The CV curves of the S@BPCS composite exhibit two reduction peaks. The peak at 2.12 V corresponds to the reduction of sulfur to long-chain Na-polysulfides ($Na_2S_X$, $X \geq 4$) and the other reduction peak at a lower potential, 1.6 V, corresponds to the further reduction of long-chain polysulfides to short-chain Na-polysulfides ($Na_2S_X$, $X \leq 4$). The anodic scan also exhibits two oxidation peaks, at 2.05 and 2.6 V, which are attributed to the oxidation of short-chain polysulfides to long-chain polysulfides and sulfur[19]. The redox processes indicate high reversibility of the cell reactions.

The GCD of the S@BPCS composite was also evaluated. Higher capacity and longer potential plateaus suggest that the BPCS sulfur host can assist as an electrocatalyst to accelerate the cathode redox reactions and enhance the utilization of active sulfur. Figure 4a shows the 1st, 2nd, and 50th charge–discharge profiles of the S@BPCS composite electrode with 4.4 mg cm$^{-2}$ mass loading and 29 μm thickness (Supplementary Fig. 16). The 1st reversible discharge capacity is 1347 mAh g$^{-1}$ at a high current density of 0.5 C and the profiles clearly show long plateaus. The plateaus at higher potentials are due to the formation of the solid–liquid state of sulfur and NaPSs, while the plateaus at lower potential are due to the reduction of long-chain NaPSs ($Na_2S_8$, $Na_2S_6$, and $Na_2S_5$) to short-chain polysulfides ($Na_2S_3$, $Na_2S_2$, and $Na_2S$). The cycling performance of the S@BPCS composite shows excellent stability (Fig. 4c), the specific capacity of 2nd discharge was 755 mAh g$^{-1}$ and after 350 cycles, it was 701 mAh g$^{-1}$ at a high current density of 0.5 C with 98.5% Coulombic efficiency. The coulombic efficiency of all the metal chalcogenide@S electrodes are given in Fig. 4f. The capacity decay per cycle is 0.0126 %, which is significantly lower than anything previously reported in the literature (Supplementary Fig. 17). It is worth noting that pure BPCS has a negligible contribution to the capacity under the same charge-discharge conditions (Supplementary Fig. 18). The cycling performance of other metal chalcogenides is shown in Supplementary Fig. 19. In addition, the Celgard polymer separator was used to test the cycle performance of BPCS@S and KB@S, and compared with the glass fiber separator in the supporting information (Supplementary Fig. 20). When using the glass fiber separator instead of the Celgard polymer separator, diffusion of polysulfides is slowed because of its thickness and porous structure, resulting in good electrochemical performance.

The rate performance of the sulfur cathodes was also measured. Figure 4b shows the rate performance of S@BPCS at different current densities from 0.5 to 3 C. The cathode S@BPCS composite exhibits capacities as high as 755, 565, 475, 415, 382 and 349 mAh g$^{-1}$ at current densities of 0.5, 1, 1.5, 2, 2.5, and 3 C, respectively. Compared to other metal chalcogenides, the S@BPCS composite has a higher rate capability, as shown in Supplementary Fig. 21. Also, when the current density was switched back to the initial value, the S@BPCS composite exhibits a high reversible capacity of 727 mAh g$^{-1}$ as compared to other metal chalcogenides (486, 432 and 399 mAh g$^{-1}$ for S@BPCO, S@BPCSE, and S@BPCTE, respectively). To the best of our knowledge, the S@BPCS composite has remarkable rate capabilities as compared with state-of-the-art materials previously reported in the Na–S battery literature, as shown in Fig. 4h[16,21,27–39].

In order to take full advantage of the high theoretical capacity of the S cathode, a high mass loading is a key factor to scale up the practical application of the Na–S battery. However, high loading of insulating sulfur will facilitate the shuttling of NaPSs and the capacity decay will be more serious with a decrease in the utilization rate of active sulfur and a decrease in the capacity of active sulfur. Therefore, it is still a great challenge to optimize the performance of high sulfur loading in Na–S batteries. Previous studies have reported that only batteries with a capacity greater than 4 mAh cm$^{-2}$ can exceed commercial LIBs[40]. In addition to this, considering that the voltage of the Na–S battery is lower than that of SIBs, a higher areal capacity is required for the former system. In this work, with the mass loading of 4.4 mg cm$^{-2}$, the capacity of the S@BPCS cathode was more than 4 mAh cm$^{-2}$. To verify the superiority of the S@BPCS cathode, batteries with high mass contents of 7.3 and 9.1 mg cm$^{-2}$ were cycled at 0.5 C (Fig. 4d, e). An initial areal capacity of 6.24 mAh cm$^{-2}$ was measured and this value stabilized to 5.5 mAh cm$^{-2}$ after 10 cycles when the mass content was 7.3 mg cm$^{-2}$. For sulfur content as high as 9.1 mg cm$^{-2}$, the initial areal capacity of the S@BPCS cathode was 8 mAh cm$^{-2}$, stabilizing at 6.6 mAh cm$^{-2}$ after 10 cycles. It is noteworthy that the current density is calculated to be as high as 9.7 mA cm$^{-2}$. Figure 4g summarizes the areal capacities of different sulfur-loaded batteries. The electrode masses of 7.3 and 9.1 mg cm$^{-2}$ used in this study are much higher than those of previously reported Na–S batteries (Supplementary Fig. 22) and commercial LIBs (4.0 mg cm$^{-2}$). To illustrate the importance of this work, we compared the capacity and rate capabilities of the Na–S batteries in Fig. 4h.

**Mechanisms of reversible reactions in the Na–S battery**. To investigate the electrochemical reactions and corresponding mechanisms, in situ XRD, in situ Raman, ex situ HRTEM/SAED, DFT calculations, and ex situ XPS analysis were performed. The in situ Raman spectrum which represents a vibration band for sulfur at 475 cm$^{-1}$[25], at a potential of 2.8 V, which is supported by the in situ XRD pattern. In Fig. 5j, the XRD pattern has a diffraction peak at 23.08°, which is indexed to the (240) plane of sulfur (JCPDF# 08-0247). The intensity of the sulfur vibrational band was reduced and a new vibrational band appears at 451 cm$^{-1}$ at a potential of 2.25 V, which is attributed to long-chain Na-polysulfides ($Na_2S_8$). This result was supported by in situ XRD, where the diffraction peak appearing at 22.95° is due to the formation of Na-polysulfides by the reduction of sulfur. After the long-chain Na-polysulfides reduced to short-chain polysulfides on further discharge, the reduction mechanism was studied. Figure 5a, b shows HRTEM/SAED images, which confirm the reduction of sulfur to Na-polysulfides ($Na_2S_5$) at a potential of 2 V. This is consistent with the XRD diffraction peak at 12.198° of $Na_2S_5$ (JCPDF# 27-0792) corresponding to the (020) plane and the Raman vibrational bands at 474 and 451 cm$^{-1}$

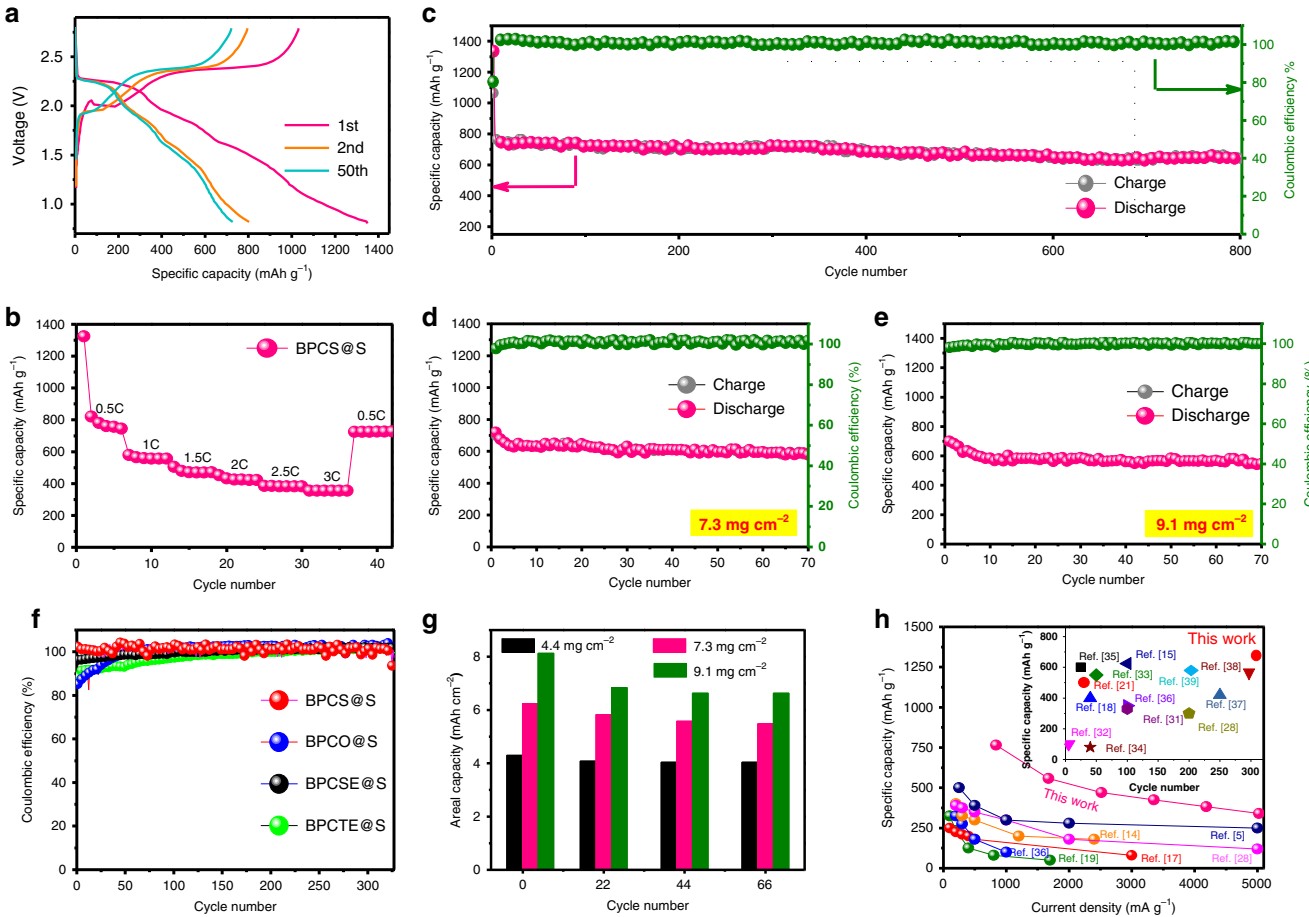

**Fig. 4 Electrochemical performance of the bipyramid prism S@BPCS electrode. a** Charge–discharge profiles at different cycle numbers. **b** Rate capabilities at different current densities. **c** Rate performance and the corresponding Coulombic efficiency at 0.5 C. **d** Rate performance and the corresponding Coulombic efficiency with mass loading 7.3 mg cm$^{-2}$. **e** Rate performance with mass loading 9.1 mg cm$^{-2}$. **f** Coulombic efficiency of different metal chalcogenides. **g** Areal capacities of all electrodes with different mass loadings at different cycle numbers and **h** comparison of the rate performance and capacity of previously reported Na–S batteries with this work.

attributed to $Na_2S_5$. On further discharge at 1.65 V, the long-chain polysulfides successfully reduced to short-chain polysulfides ($Na_2S_4$) and the corresponding

XRD pattern shows a diffraction peak at 19.93° indexed to the (112) plane that was strongly attributed to $Na_2S_4$ (JCPDF# 25-1112) and it was robustly confirmed with HRTEM/SAED (Fig. 5c, d) and a Raman vibrational band at 472 cm$^{-1}$[25]. Furthermore, for discharge at 1.2 V, a new XRD diffraction peak was generated at 11.84° which is indexed to the (101) plane of $Na_2S_2$ (JCPDF#81-1764) and was subsequently confirmed with HRTEM/SAED (Fig. 5e, f) and a Raman vibrational band at 474 cm$^{-1}$ for $Na_2S_2$[25]. Subsequently, when the cell was deeply discharged at 0.8 V, a new peak at 16.5° was generated which indexed to the (122) plane of JCPDF#47-0178 and is ascribed to $Na_2S$[41]. Figure 5i shows the CV curve labeled with polysulfide formed at different voltage during discharge process. Figure 5g, h showing HRTEM/SAED and a Raman vibrational band (Fig. 6b) at 473 cm$^{-1}$ which is attributed to $Na_2S$, supporting the XRD result. The first discharge process could, therefore, be labeled with corresponding polysulfides as described in Fig. 6a, and the entire reduction process can be written as follows:

$$S \rightarrow Na_2S_X \rightarrow Na_2S_{6,5} \rightarrow Na_2S_{4,3} \rightarrow Na_2S_2 \rightarrow Na_2S$$

When the cell was charged to 2.8 V, $Na_2S$ and $Na_2S_5$ are not detected in the XRD pattern. This means that the conversion rate of short-chain polysulfides to long-chain polysulfides is fast, and

that could be a reason why $Na_2S$ and $Na_2S_5$ are not detected within situ XRD. In addition, the intensity of the XRD peaks decreases, which indicates a fast transformation of S to long-chain Na-polysulfides and to $Na_2S$. This study elucidates the reduction reactions of S to $Na_2S$ and presents a new mechanism to explore the significance of polar and catalytic BPCS. The polar and catalytic BPCS diminishes the dissolution of polysulfides, facilitates the fast transformation of long-chain polysulfides to $Na_2S$, alleviates the shuttle effect, and delivers excellent electrochemical performance. Furthermore, to check for dissolution of polysulfides in the electrolyte, the fiberglass disk that was used as a separator was removed from the cell after cycling for ex situ XPS of S. Figure 6c ($c_1$–$c_3$) shows high-resolution XPS spectra of S for the S@BPCTE, S@BPCSE, and S@BPCS batteries separators, respectively. The peaks in the range of binding energies of 160–165 eV are due to dissolved polysulfides[27] and the peaks in the range of 165–170 eV are due to the formation of $SO_X$. Figure 6c ($c_3$) shows that there are no Na-polysulfide peaks in S@BPCS separator and the rest have dissolved polysulfide peaks. These findings clearly demonstrate that the polar and catalytic BPCS can efficiently capture the Na-polysulfides.

Lastly, the visible difference in the adsorption of polysulfides in the BPCS was performed using a $Na_2S_6$ solution. Figure 6d clearly shows that when carbon was added to the $Na_2S_6$ solution, there was no change in the yellow color of polysulfides whereas, when BPCS was added to $Na_2S_6$ solution, the yellowish color turns

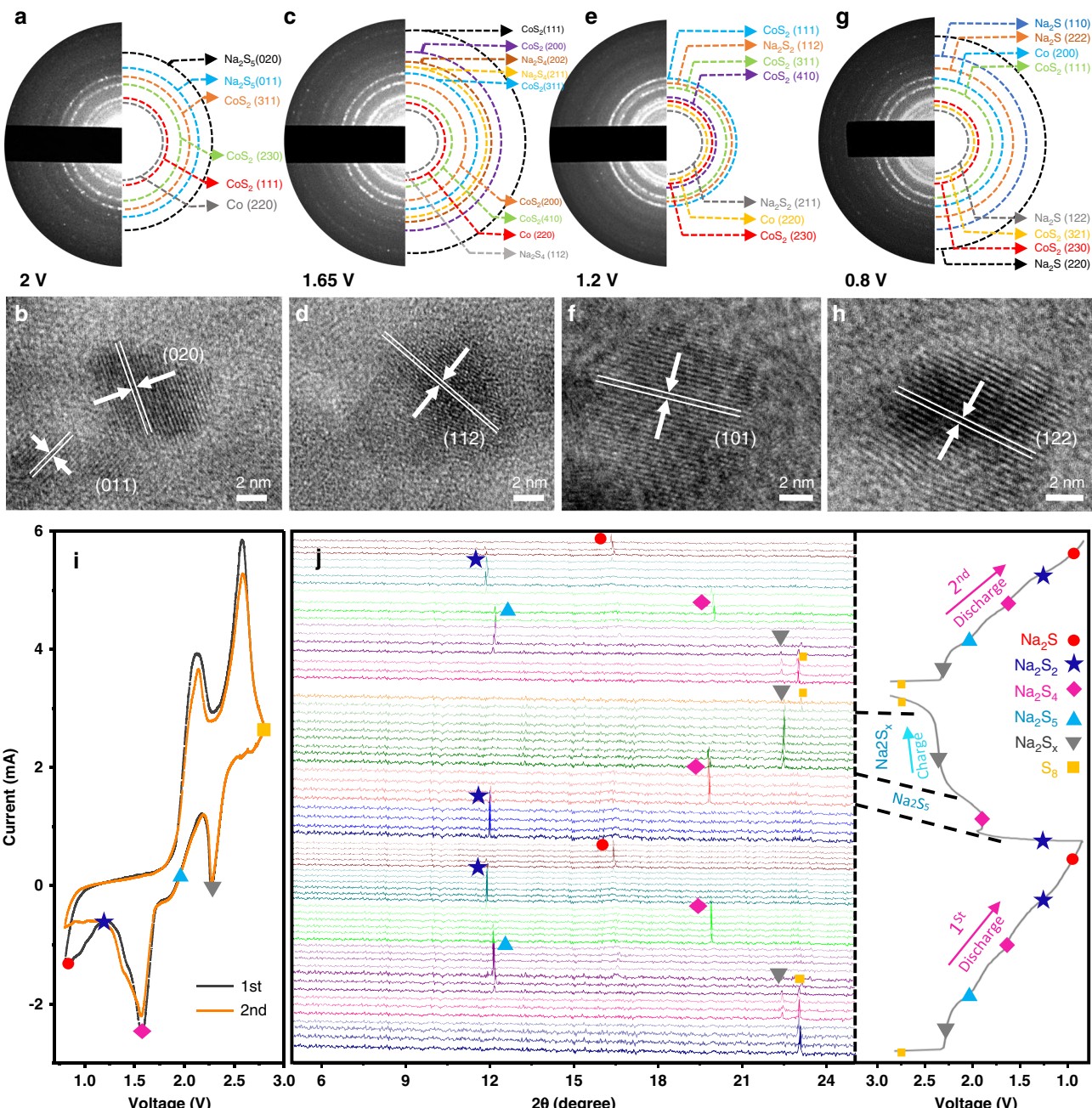

**Fig. 5 HRTEM images and SAED patterns of S@BPCS electrode at different discharge voltages. a** SAED pattern and **b** corresponding HRTEM at 2 V. **c** SAED pattern and **d** corresponding HRTEM at 1.65 V; **e** SAED pattern and **f** corresponding HRTEM at 1.2 V and **g** SAED pattern and **h** corresponding HRTEM at 0.8 V; mechanistic characterization of the S@BPCS electrode. **i** Cyclic voltammogram. **j** In situ X-rays diffraction (XRD) patters in the selected range of degree 2-theta (right) with first two charge–discharge profiles at a current density of 0.5 C.

colorless after few hours demonstrating the strong interaction and adsorption capability of polar BPCS towards NaPSs. The S@BPCS composite has internal void space of about 376 nm, which is enough to provide adequate space for NaPSs and tolerate the strain during sodiation and desodiation as shown in Fig. 6e.

**Computational results**. DFT calculations were performed with the SCAN + rvv10 functional to understand the interactions of NaPSs with $CoX_2$ cathode hosts. The lattice parameters of the structures optimized with SCAN + rvv10 are in excellent agreement with experimental results, as shown in Supplementary Fig. 23 and Supplementary Table 1. As can be seen from the density of states plots in Supplementary Fig. 24, there are states at

the Fermi level in the spin up-channel for all structures resulting in half-metallic behavior. The binding of polysulfides to the surface of $CoX_2$ materials were further investigated with slab calculations. The energies of different surface terminations of pyrite (p-$CoS_2$) and marcasite (m-$CoTe_2$, m-$CoSe_2$) $CoX_2$ structures are shown in Supplementary Table 2. The surface energies ($\gamma$) were calculated using Eq. (1)

$$\gamma = \frac{E_{slab} - N^* E_{bulk}}{2A},\qquad(1)$$

where $E_{slab}$ is the slab energy for a given surface (single point or relaxed surface), $E_{bulk}$ is the energy of relaxed bulk geometry per formula unit, $N$ is the number of formula units in the slab, and $A$

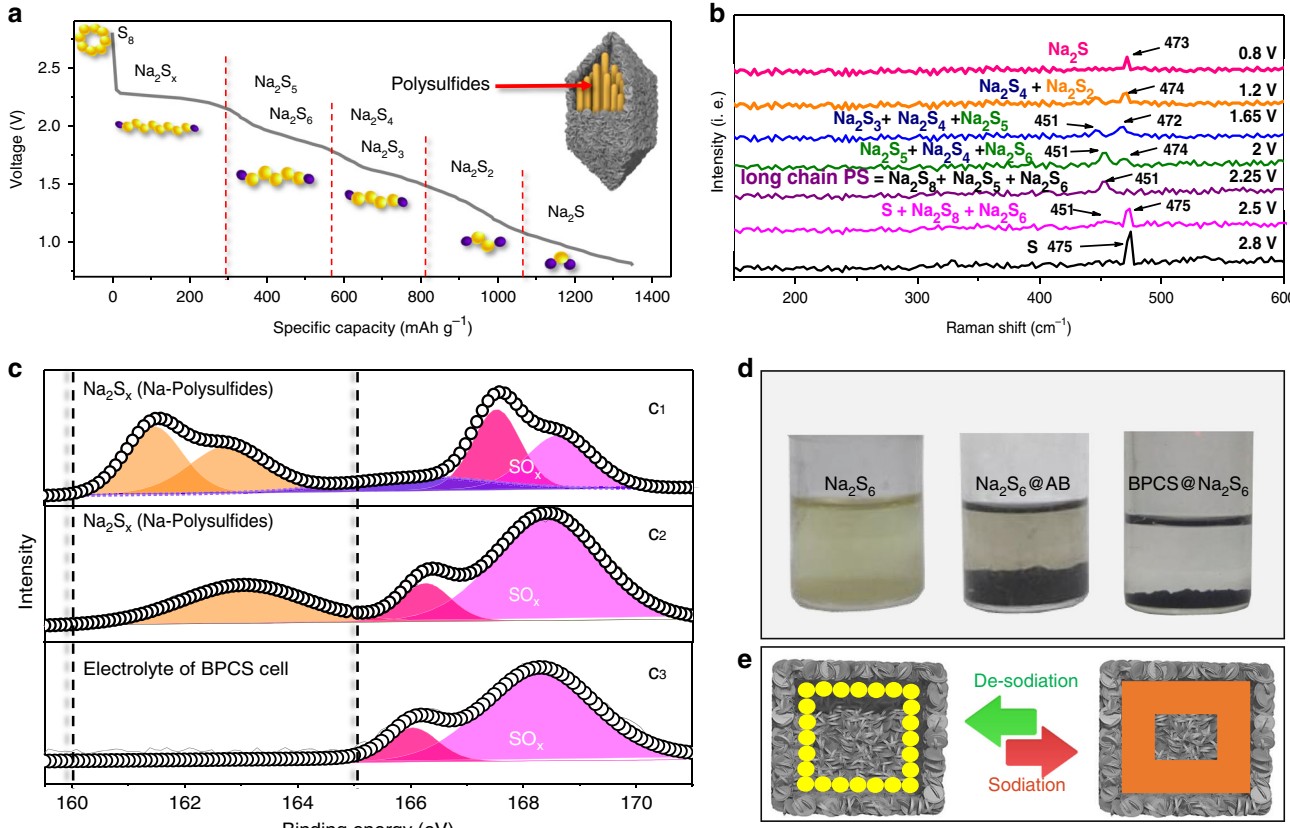

**Fig. 6 Mechanistic illustration of formation and adsorption of polysulfides. a** Initial discharge of the S@BPCS composite labeled with practically interpreted polysulfides at different potentials. **b** In situ Raman spectra of the S@BPCS cell at different potentials. **c** XPS spectra of the S 2p of S@BPCS cell electrolyte after cycling. **d** Visual display of adsorption of $Na_2S_6$ by BPCS and **e** scheme presenting the volume change during sodiation and desodiation.

is the area of the surface. To screen a wider range of possible surfaces for different $CoX_2$ structures, the cheaper PBEsol functional was used instead of the SCAN + rvv10 functional. Supplementary Table 2 shows the surface energy of the eight-lowest energy surfaces of p-$CoS_2$ and the ten lowest energy surfaces of m-$CoSe_2$ and m-$CoTe_2$. As can be seen from Supplementary Table 2, the lowest energy surface for p-$CoS_2$ is (100), followed by the (210) surface, which is consistent with previous XRD and TEM results[28,29]. Several low energy surfaces are present in the m-$CoSe_2$ and m-$CoTe_2$ marcasite structures, including the (010), (101), and (211) surfaces, which is consistent with low energy surfaces observed computationally and experimentally for analogous marcasite surfaces[30,42]. The (100) surface of p-$CoS_2$ and the (101) and (010) surfaces of m-$CoSe_2$ and m-$CoTe_2$ were chosen to study the binding of polysulfides. The geometries of all of the low energy slabs predicted with PBEsol were reoptimized with SCAN + rvv10 before further binding calculations.

In order to understand the nature of binding between NaPSs and the cathode hosts, the enthalpy of $Na_2S_y$ polysulfide molecules of different chain lengths ($y$ = 1, 2, 4, 5, and 6) bound to both the p-$CoS_2$ (100) surface and a graphene sheet were calculated. The formation energy of the NaPSs molecules was taken as the difference in the energy between a slab of the cathode host with ($E_{slab+NaxS}$) and without the NaPSs ($E_{slab}$), referenced to the solid S and $Na_2S$ phases via $E_{form} = E_{slab+NaxS} - E_{slab} - x/2 \times E_{Na2S} - (1 - x/2) \times E_s$. The effects of solvation on the slabs with and without the bound NaPSs were taken into account via the inclusion of an implicit solvation model in VASPsol[43]. The lowest energy binding geometries for each NaPS are shown in (Supplementary Fig. 25). The formation energies ($E_{form}$) of solid phases of $Na_2S$, $Na_2S_2$, $Na_2S_4$, $Na_2S_5$, and $Na_2S_6$, and S were calculated

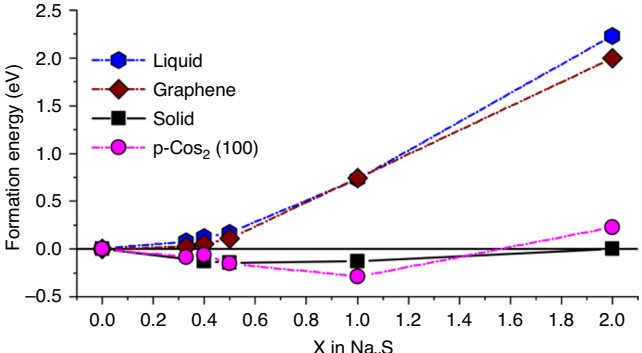

**Fig. 7 Convex energy hull of $Na_2S_y$ phases per mol of S ($x = 2/y$) from DFT calculations.** Energy of solid $Na_2S_y$ structures is shown with black symbols relative to end-member compositions of solid $Na_2S$ and S. The energies of molecular $Na_2S_y$ geometries bound to the p-$CoS_2$ (100) surface (magenta), graphene surface (brown) and in the free liquid state (blue) with implicit solvation are shown with dashed lines.

using the SCAN + rvv10 functional as shown in Fig. 7 (black symbols). The formation energies of intermediate $Na_2S_y$ phases ($E_{NaxS}$) normalised to one mol of S ($x = 2/y$) are reported relative to the end member configurations of $Na_2S$ ($E_{Na2S}$) and S ($E_s$) by the formula $E_{form} = E_{NaxS} - x/2 \times E_{Na2S} - (1 - x/2) \times E_s$. With respect to Na metal, the sequence of reactions from S → $Na_2S_5$ → $Na_2S_4$ → $Na_2S_2$ → $Na_2S$ are predicted to occur at voltages of 2.28 V → 2.08 V → 1.91 V → 1.82 V, which are in good agreement with previous computational studies[44] and the experimental cycling data in Fig. 6a, particularly for the reactions

from S to $Na_2S_4$, where a significant overpotential does not occur. The formation energy of the free $Na_2S_y$ molecules in the presence of the implicit solvent, but without the presence of the catalyst host, is also shown in Fig. 7 to approximate the energy of the "liquid" state. From Fig. 7, it can be seen that the formation energy of the NaPSs bound to the graphene cathode host is very similar to the free NaPSs in the "liquid", indicating a very weak binding between the nonpolar graphene surface and the NaPSs as observed in previous studies[45].

In comparison, the formation energy of all NaPSs bound to the p-$CoS_2$ (100) surface are lowered significantly, primarily due to the strong interaction between the $Na_2S_y$ molecules and the polar p-$CoS_2$ surface. The difference in the energy between the NaPSs bound to the p-$CoS_2$ (100) surface and the 'liquid state' increases systematically as the chain length, $y$, decreases ($x$ increases), reaching 2 eV for the $Na_2S$ molecule. As discussed further in Supplementary Fig. 26a, the strong binding of the NaPSs to the surface is related to charge transfer between the undercoordinated $Co^{2+}$ sites on the surface to the S atoms in the $Na_2S_y$ cluster.

Interestingly, after binding to the p-$CoS_2$ (100) surface, the enthalpy of the bound $Na_2S_y$ molecules closely matches the energy of the solid $Na_2S_y$ phases. An absolute comparison of the energies is not possible from Fig. 7 as entropic contributions and the presence of explicit solvation have not been included, however, this close matching between the energy of the molecular NaPSs and solid $Na_2S_y$ phases is likely to be beneficial, in which reactions in the molecular state are expected to be facile. For example, if the binding of a $Na_2S$ molecule to the cathode host is weak, as in the graphene case, the voltage required to drive the reaction $Na_2S_2 + 2Na^+ + 2e^- \rightarrow 2Na_2S$ on the graphene at is 0.59 V. As the voltage for the solid–solid reaction of $Na_2S_2$ to $Na_2S$ is 1.82 V, a large overpotential of 1.23 V would be required to facilitate the reaction in the molecular state. If the cathode host binds the NaPSs too strongly, then the energy of the anchored $Na_2S$ molecule will be lower than the $Na_2S$ solid. $Na_2S$ molecules will therefore cover the cathode host surface instead of forming bulk $Na_2S$ particles, reducing the ability for it to catalyze further reactions. By balancing the energy between the anchored and bulk $Na_2S_y$ materials, facile molecular reactions can take place on the cathode host surface which lead to the formation of bulk $Na_2S$ particles.

The formation energy of $Na_2S_y$ molecules bound to the marcasite p-$CoSe_2$ (100), m-$CoSe_2$ (010), m-$CoSe_2$ (101), and m-$CoTe_2$ (010) surfaces were also calculated, as shown in Supplementary Fig. 27. It can be seen from Supplementary Fig. 27 that all of the $CoX_2$ hosts strongly bind the NaPS molecules, resulting in formation energies that are close to those of the solid $Na_2S_y$ phases. For the $Na_2S$ end member, a small difference in the formation energy is observed on analogous surfaces of materials with the same structure, but with a different anion, i.e., p-$CoS_2$(100) vs. p-$CoSe_2$(100) and m-$CoSe_2$(010) vs. m-$CoTe_2$(010). The difference in the formation energy of a $Na_2S$ molecule is larger on different surfaces of the same structure, e.g., m-$CoSe_2$(101) vs. m-$CoSe_2$(010), suggesting that structural differences in these materials (pyrite vs marcasite) may play a more important role in this family of materials than the nature of the anion (i.e., S, Se, and Te). This result is in line with the experimental electrochemical results in Fig. S20, where the marcasite based BPCSE and BPCTE structures have similar rate performance, whereas the pyrite based BPCS structure has markedly better rate performance.

## Discussion

Hollow polar and catalytic bipyramid prism $CoS_2$/C (BPCS) have been designed as an efficient sulfur host for Na–S batteries. BPCS have a unique architecture with a hierarchical surface and wide internal spaces that provide sufficient room to accommodate sodium polysulfides (NaPSs) and can withstand volume expansion during sodiation and desodiation. This study provides a systematic method for further understanding of the reversible reaction mechanism during the charge/discharge process. In/ ex situ experimental results elucidate the discharge mechanism and confirm the importance of polar and catalytic BPCS in accelerating the electrochemical reaction and enabling the direct conversion of short-chain polysulfides to long-chain polysulfides (instead of $Na_2S_5$) without affecting the reactants or products, thereby reducing the one-step reaction process and accelerating the reaction kinetics. Furthermore, DFT calculations support the mechanism that polysulfide adsorption is superior for metal sulfides, selenides and tellurides with interwoven surfaces and chemical composition when compared to carbon hosts. This study shows that a polar and catalytic sulfur host with unique architecture can catalyze the polysulfides conversion reactions and suppress the shuttle effect by chemisorption, resulting in good electrochemical performance. While the focus of this study was on cobalt based catalytic sulfur hosts, the combined experimental and computational methodology used in this work is widely applicable to other transition metal dichalcogenide systems.

## Methods

**Synthesis of metal chalcogenides host and their sulfur composites**. *Synthesis of Co-bipyramid prisms:* Typically, 1.0 g of polyvinylpyrrolidone (PVP) and 0.135 g of cobalt acetate tetrahydrate were dissolved in 500 ml of ethanol at room temperature under sonication to form a clear solution. The solution was then heated to 90 °C under reflux conditions for 1.5 h. Afterward, the precipitates were collected via centrifugation at 2795 × g, rinsed thoroughly with hot ethanol and water (preheated at 60 °C) for at least 15 times to remove the attached PVP on the surface before being fully dried in air (Fig. 1a).

*Synthesis of Co/C hollow bipyramid prisms:* To synthesize hollow bipyramid prisms, the co-precursor was calcined at 500 °C for 3 h with the ramp rate of 5 °C min⁻¹ under Ar and $H_2$ mixed atmosphere.

*Synthesis of hollow $Co_3O_4$/C bipyramid prisms with the hierarchical surface (BPCO):* To synthesize hollow BPCO the above Co/C was calcined at 550 °C for 2 h with the ramp rate of 3 °C min⁻¹ under an air atmosphere.

*Synthesis of $CoSe_2$/C bipyramid prisms with the hierarchical surface (BPCSE):* To synthesize BPCSE, the BPCO was calcined with Se powder putting in separate boats according to up and low stream at 300 °C for 3 h with the ramp rate of 2 °C min⁻¹ under Ar atmosphere.

*Synthesis of $CoS_2$/C bipyramid prisms with the hierarchical surface (BPCS):* To synthesize BPCS, the BPCO was refluxed with thioacetamide (TAA) for 12 min at 90 °C and precipitates were collected with centrifuge machine, after drying, the powder was heated at 350 °C for 2 h with the ramp rate of 2 °C min⁻¹ under Ar atmosphere.

*Synthesis of $CoTe_2$/C bipyramid prisms with hierarchical surface (BPCTE):* To synthesize BPCTE, the BPCO was heated with Te powder, putting it at low stream in separate boats and Te powder at upstream at 700 °C for 3 h with the ramp rate of 3 °C min⁻¹ under Ar atmosphere.

*Synthesis of S@BPCS:* For the synthesis of sulfur cathode, the sulfur host and sublimed sulfur were well mixed in mortar with the weight ratio of 1:4 and then transferred to a heating boat covered with Al-foil and heated at 155 °C in a tube furnace for 12 h with a ramp rate of 2 °C min⁻¹ under an Ar atmosphere. Additionally, the sulfur composite was heated at 190 °C for 30 min. to remove the excessive sulfur from the surface. Sulfur composites of all other metal chalcogenides were also prepared by the similar method.

*Synthesis of $Na_2S_6$ solution:* For the synthesis of $Na_2S_6$, $Na_2S$ and S were stirred in tetraethylene glycol dimethyl ether at 80 °C for 12 h. The dark yellowish black colour $Na_2S_6$ solution was obtained with 0.067 M concentration, which was further diluted before the absorption test.

**Materials characterization**. The morphological investigations were done by FESEM (JSM-7800F, Japan), SEM (JSM-6510LV) and TEM (JEM-2100, Japan). The elemental composition was measured by EDS using EDS, JEOL-6300F. Phase purity and chemical compositions of all the samples were investigated by XRD (MAXima-X XRD-7000) in the range of 5–90° at a scan rate of 5° min⁻¹ and by XPS was using a Thermo Scientific ESCALAB 250Xi electron spectrometer. TGA was performed to measure the sulfur contents in the host using an SDTQ600 analyzer (TA instrument) in the Ar atmosphere at 600 °C with a heating rate of 10 °C min⁻¹. The textural properties (pore size and surface area) were investigated by nitrogen adsorption–desorption using BET (Quantachrome Instruments, USA).

**In situ characterization**. The in situ Raman spectra were collected using Lab-RAM HR Evolution (Horiba) Raman microscope (EL-CELL Germany), with excitation wavelengths laser of 532 nm under discharged/charged at a sweep rate of 1 mV s$^{-1}$ using CHI workstation (CHI 660D). For in situ XRD, the preparation method of half-cell was similar, 4 mm hole was punched on the steel caps and steel spacer on both the anode and cathode sides and the holes were completely sealed with Kapton film and an AB glue coating. To enhance the intensity of XRD peaks, a thicker layer of cathode material was deposited (7 mg cm$^{-2}$) on an aluminum substrate. The discharge/charge was performed on a LAND workstation (Wuhan, China) at a current density of 500 mA g$^{-1}$.

**Ex situ characterization**. The ex situ HRTEM and SAED patterns were obtained at different stages of discharge. Firstly, batteries were discharged up to the required potential using a LAND workstation and at the end of discharge, the batteries were opened in a glove box to collect sample material. Afterward, the samples were further used for TEM measurements. For ex situ XPS, the batteries were discharged up to the mentioned potential and after batteries were opened in the glove box and collected sample used for XPS analysis. The ex situ XPS of battery electrolyte was also performed after full discharge at 0.8 V.

**Electrochemical measurements**. The electrochemical measurements were performed using CR2032 coin-type cells. Films of the active materials were formed from a slurry of active material (cathode material), acetylene black (AB) carbon and polyvinylidene difluoride, mixed in the ratio of 8:1:1 in N-methyl-2-pyrrolidone solvent. After mixing, the slurry was spread on a circular disk of aluminum with a doctor blade and dried at 60 °C for 12 h. CR2032 coin cells were assembled in an Ar filled glove box, pure sodium foil was used as a counter and a reference electrode and a Whatmann glass fiber disk was used as a separator with 1 M NaClO$_4$ in tetraethylene glycol dimethyl ether as an electrolyte. The separator was used after drying at 120 °C for 12 h. The long cycle performance and rate capabilities were performed on LAND (Wuhan, China) electrochemical workstation in the potential window of 0.8–2.8 V and CV was performed at Arbin electrochemical workstation at a scan rate of 0.1 mV s$^{-1}$.

**DFT calculations**. DFT calculations were performed to understand the interactions of NaPSs with CoX$_2$ cathode hosts. Spin-unrestricted DFT calculations were performed in the VASP code using projector augmented wave pseudopotentials[46,47]. The SCAN meta-GGA functional has been shown to give accurate structures and energetics for a range of diversely bonded solid and molecular systems[48,49]. In this study, we used the recently developed SCAN + rvv10[50,51] extension to the original SCAN functional, which provides an accurate description of short, intermediate and long-range van der Waals interactions which are known to play an important role in binding in transition metal sulfide materials[45]. Bulk phases of CoS$_2$ and CoTe$_2$ were modeled with the pyrite (Pa-3) and marcasite (Pnnm) structures, respectively, and will subsequently be referred to as p-CoS$_2$ and m-CoTe$_2$. In this study, the CoSe$_2$ bipyramidal host primarily adopted the marcasite structure (m-CoSe$_2$). However, previous studies have shown that the energies of the pyrite and marcasite structures of CoSe$_2$ are similar and so both the pyrite (p-CoSe$_2$) and m-CoSe$_2$ phases were considered in this study for comparison. Additional details of the DFT calculations are presented in Supplementary Note 1.

## Data availability
The data that supports this work is available in this paper, including supporting information and response files.

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

## Acknowledgements
We appreciate the support from the National Natural Science Foundation of China (Nos. 21773188 and 21972111), Fundamental Research Funds for the Central Universities (XDJK2019AA002), Postgraduate tutor team-building project (XYDS201911). The work at UT was supported by the Welch Foundation (F-1841) and the Texas Advanced Computing Center.

## Author contributions
M.K. Aslam designed the project, conducting synthesis, and electrochemical tests. I.D. Seymour and N. Katyal did the DFT calculations. S. Li helped with the figures. T.T. Yang carried out the thermogravimetry experiments and purchased chemicals. G. Henkelman supervised the DFT calculations. S.J. Bao and M.W. Xu jointly supervised the project.

## Competing interests
The authors declare no competing interests.
