## [Peer Review File · Nature Communications]

REVIEWER COMMENTS

Reviewer #1 (Remarks to the Author):

This manuscript reports Na-S batteries with metal chalcogenides as sulfur hosts. The performance is very promising and results are well studied using DFT and in situ measurements. Some comments need to be addressed before publication:

Pore size and surface area measured by BET are not clearly stated in the text. Is the structure micro/meso/macroporous?

Exact values of coulombic efficiency should be provided (instead of saying almost 100%).

A Whatman glass fiber separator is used instead of a typical Celgard polymer separator. The former is much thicker and will reduce energy density. What is the cycling performance using Celgard separator?

350 cycles is quite short for cycling. What is the performance like after 350 cycles and how can it be improved?

Words in the figures are way too small.

Supplementary figures are not numbered sequentially, making it hard to follow.

Some relevant papers on sodium anodes should be referenced: Adv. Energy Mater. 2017, 7, 1602528 and ACS Cent. Sci. 2015, 1, 449-455.

Some recent papers on Na-S batteries are also missing and should be included in the comparison in Fig 4h: Cell Rep. Phys. Sci. 2020, 1, 100044 and Energy Storage Mater. 2020, 29, 1-8.

Reviewer #2 (Remarks to the Author):

Aslam et al. report here a new host material for sulfur batteries, namely sodium sulfur batteries.

The material is prepared by carbonizing a Co acetate crystal which in effect gives a Co/C material and this serves as sulfur host.

There are many reports on sulfur host materials and it is currently known that anode and electrolyte are more of a problem in these battery systems than new cathode host materials. And it is always a bit suspicious if the one and only solution comes up with a new quite exotic composite material.

The authors claim that encapsulation and catalytic effects are important here. From the data given it is quite difficult to judge where the shuttle suppression comes from.

The surface area of the material is quite low, so adsorption alone cannot be the mechanism. The material is not an enclosure (sulfur can access the inner core, polysulfides can go in and out). It might be a good idea to quantify the adsorption capacity with UV Vis following the method reported by Hippauf. This would give a good comparison with other more polar carbon host materials as reported by Hao. Also a water isotherm would be helpful to check the hydrophilic character of the host.

Of course a big flaw are the limited experimental details. No electrolyte/sulfur ratio given. That's a step backwards in reporting accurate data. With a whatman separator (thick) an excess of electrolyte is guaranteed! The huge capacity loss in the first cycle confirms this view (dissolution). And in a flooded cell PS diffusion takes longer. And the glass fibres also adsorb the PS, and affect crystallization.

It would be good to test the material also with a PE separator and compare it to a Ketjen Black based reference cathode. And check performance also under lean conditions with E/S ratios varying between 5 and 10.

The in situ analysis is interesting. However, do the authors think these solid phase intermediates take part in the conversion mechanism, or are just side products? The catalytic effect is probably only important for dissolved species.

A positive aspect is the massive materials characterization via SEM, XPS etc.

In summary it might be worth reconsidering this work if the material also performs well at low E/S ratios with a PE separator.

Additional points:

1) Severe technical problem: adsorption data and porosity analysis

- This material has no high surface area, as one can see from the N₂ isotherms (S8)
- Fig. S8 probably shows a N₂ adsorption isotherm at 77 K, but is labeled „surface area“
- Hence, the author is not familiar with this technique
- Moreover: The isotherm does not close, probably not enough mass used for measurement
- ♦ repeat (BPCS)
- Pore size analysis is meaningless

2) The bipyramidal aspect is overemphasized. The crystals look almost like prism. Does the shape really play a role?

3) Conversions are nicely depicted in the supportings. This scheme would be helpful for the reader in the main text.

4) XRD: sulfur, alpha or beta?

5) What is TAA?

Reviewer #3 (Remarks to the Author):

My review is for the DFT section of this submission, but I read the entire manuscript to understand the context of the DFT simulations. The results presented on a new polar bipyramid prism CoS₂/C sulfur host (cathode) for sulfur batteries seem very promising and worthwhile of publication in Nature Communications, if the experiments hold up to scrutiny by experimental experts. With some edits, I support the publication of the DFT sections, if the other reviewers endorse the experimental sections.

The only two general comments I have are 1) on the “catalytic” description of the BPCS. I think a description of why the material is described as catalytic is warranted near the beginning of the paper, early in the introduction, because otherwise I am unsure why this description is so necessary. Don't all cathodes “catalyze” reactions?

2) Why is cobalt used rather than another metal; is the choice of cobalt important? I don't think the choice of cobalt versus other transition metals is discussed in the paper and the computational authors could test other metal ions!

My general comments on the DFT section is that it's too long. For example, the section on the charge density differences (Fig 7b) are unsurprising and not that informative; the magnetic moment calculation may not be of interest to a general audience. Some of the DFT section should be moved to the supplemental information because the gas-phase simulations may not be such a good proxy for the real reactions in the electrolyte, 1M NaClO₄ in tetra-ethylene glycol dimethyl ether. For example, the charge density differences could look at lot different if explicit solvent was

included.

Specific comments:

In the discussion section, line 438, the authors state, "DFT calculations support the mechanism that sulfide adsorption is superior for homogenous metal sulfides...when compared to the carbon host." Unless I'm mistaken, this is one of the first times "homogenous" is used to describe the surfaces and it's unclear what the authors mean by "homogeneous". Otherwise, the statement seems well backed up by their calculations.

The authors do not comment on using a Hubbard-U term to correctly capture the electronic structure due to the cobalt ions. XPS of the cobalt has been performed (Figure 3), so a comment would be beneficial. Also, since the magnetic moment is described in detail, the effect of a Hubbard U should be discussed.

The authors do a thorough search for the low energy surfaces of CoX₂ materials to create slab models. The authors do not clearly comment on why CoS₂ is chosen over CoSe₂ and CoTe₂ for making the BPCS. Is the sulfur in the cathode necessary for good performance? What is the take-away message from doing the selenium and tellurium calculations?

How do the authors ensure that they've left enough vacuum space to not have finite-size effect errors? Especially with the dipoles created on the surface due to the NaPSs molecules, 15 Å MAY NOT BE ENOUGH.

Line 361: It is interesting that the author choose to calculate the formation of free Na₂S₇ molecules in the presence of an implicit solvent using VASPsol. They should include an implicit solvent for the formation energies on graphene and p-CoS₂ as well, or explain why they did not do these calculations. Figure 7 would be much more convincing and clear if implicit solvent were included as a comparison. Currently, the point of Figure 7 is confusing. There is a positive formation energy in the liquid and on graphene – indicating that the compounds should not form. Some of the formation energies on p-CoS₂ are quite small – would they be positive if solvent was included?

Since trends are reported, it is acceptable, but not the best possible results, that Hubbard-U and implicit solvent are not considered when calculating the formation energy for formation on CoS₂.

Line 381: I do not see where the following result is shown in a plot or the value of binding energy explained, "The binding energy of NaPSs on the p-CO₂ (100) surface increases systematically as the chain length decreases, reaching 2eV for the Na₂S molecule." It is certainly not clear from Fig. 7 a).

Response to Reviewers

Reviewers' comments and our Response:

Reviewer #1 (Comments to the Author): =====

The performance is very promising and results are well studied using DFT and in situ measurements. Some comments need to be addressed before publication.

Response. Thank you very much for your affirmation of our work and your favorable comment on the manuscript. We are grateful for the reviewer's positive and constructive comments. All requested revisions based on the professional comments from the reviewer have been carried out.

Specific comments:

Q1. Pore size and surface area measured by BET are not clearly stated in the text. Is the structure micro/meso/macroporous?

A1. Thank you very much for your valuable constructive comment. We have provided a clear statement for BET measurement in the main text.

"...The Brunauer-Emmett-Teller (BET) surface area of BPCS is $158 \text{ m}^2 \text{ g}^{-1}$ and the hysteresis loop is of type H4 indicates the physisorption isotherm of type I, which conforms to IUPAC, demonstrating that BPCS is composed of uniform slit-like pores, representing microporous structure, as shown in Figure S13 in supporting information...(see Page 8 Line 16)"

Q2. Exact values of Coulombic efficiency should be provided (instead of saying almost 100%).

A2. We appreciate the reviewer for the very instructive suggestion. The exact Coulombic

efficiency value is 98.5 %. We have provided it in the main text. ... ” (see Page 10 Line 25)

Q3. A Whatman glass fiber separator is used instead of a typical Celgard polymer separator. The former is much thicker and will reduce energy density. What is the cycling performance using Celgard separator?

A3. Thank you very much for the very professional comment. There is some reason to use glass fiber as a separator: 1) Compared with the polymer separator, the thermal stability and mechanical properties of the glass fiber have been greatly improved. Therefore, the safety performance of sodium-sulfur battery can be greatly improved by using glass fiber as a new separator. 2) Luo et al.¹ demonstrate that the DSC and TG curves of the glass fiber fabric are flat and smooth, and there is no obvious peak valley. It shows that there is no weight change and heat absorption and release in the experimental temperature range. However, the DSC curve of commercial separator has two endothermic peaks at 280 °C and 380 °C. The peak value around 280 °C is very sharp and obvious, corresponding to the sharp decline of TG curve. This huge weight loss may be related to the decomposition of polymer at high temperature. 3) The contact angle of the Celgard polymer separator is very high, showing a negligible wettability during electrolyte filling, while the glass fiber has a zero contact angle (Figure 1) meaning that the glass fiber has a high wettability for electrolyte during filling and the charge/discharge processes.

Figure 1. Digital images of contact angle of (a) Celgard polymer separator and (b) glass fiber separator, using liquid electrolyte.

4) The glass fiber can also act as a cushion to distribute the load across the electrode surface. The Celgard usually has a denser structure and does not stick to the electrode surface as easily. This helps for post-mortem analysis of cells. 5) When using the glass fiber separator instead of the Celgard polymer separator, diffusion of polysulfides is slowed because of its thickness and porous structure, resulting in good electrochemical performance as shown in Figure 2 and in supporting information Figure S20.

Figure 2. Cycling performance of BPCS@S and Ketjen Black@S composites using glass fiber and Celgard polymer as separator.

Figure 2 shows the clear performance comparison of glass fiber and Celgard polymer separator, as well as performance comparison of BPCS@S and KB@S composites.

In summary, the glass fiber separator has high porosity and wettability, improves the absorption of dissolved polysulfide, slows down the diffusion process, and has good electrochemical performance.²

Q4. 350 cycles is quite short for cycling. What is the performance like after 350 cycles and how can it be improved?

A4. We have provided long cycling performance of BPCS@S in the main text (Figure 4c). The specific capacity gradually decreases from 751 to 675 after 800 cycles at 0.5 C, as shown in Figure 3. This is good cycle performance and the capacity attenuation rate is of 0.0126% per cycle.

Figure 3. Long cycling performance of BPCS@S at 0.5 C.

Q5. Words in the figures are too small.

A5. We have modified all possible figure labels.

Q6: Supplementary figures are not numbered sequentially, making it hard to follow.

A6. We are sorry for the ambiguity. We have revised the figure numbers in the supplementary information.

Q7. Some relevant papers on sodium anodes should be referenced: Adv. Energy Mater. 2017,

7, 1602528 and ACS Cent. Sci. 2015, 1, 449-455.

A7. Thanks very much for your kind comments. We have now cited these papers as Ref. 6 and Ref. 7 in the revised manuscript.

Q8. Some recent papers on Na-S batteries are also missing and should be included in the comparison in Fig 4h: Cell Rep. Phys. Sci. 2020, 1, 100044 and Energy Storage Mater. 2020, 29, 1-8.

A8. We have included these papers in performance comparison in the main text, cited as Ref. 50 and Ref.51 in the revised manuscript.

Reviewer #2 (Comments to the Author): =====

Q1. There are many reports on sulfur host materials and it is currently known that anode and electrolyte are more of a problem in these battery systems than new cathode host materials. And it is always a bit suspicious if the one and only solution comes up with a new quite exotic composite material. The authors claim that encapsulation and catalytic effects are important here. From the data given it is quite difficult to judge where the shuttle suppression comes from.

The surface area of the material is quite low, so adsorption alone cannot be the mechanism. The material is not an enclosure (sulfur can access the inner core, polysulfides can go in and out). It might be a good idea to quantify the adsorption capacity with UV Vis following the method reported by Hippauf. This would give a good comparison with other more polar carbon host materials as reported by Hao. In addition, a water isotherm would be helpful to check the hydrophilic character of the host.

A1. Thank you very much for your professional and valuable comments. No doubt, there are several main problems of the rechargeable Na-S battery which hinder its practical application. Sulfur exists in the form of polyatomic molecules with different structures. Cyclo-S₈ is the

most stable allotrope at room temperature. In the process of charging and discharging, cyclo-S₈ experienced a series of structural and morphological changes, forming soluble polysulfide Na₂S_x (8 ≤ X ≤ 3) and insoluble sulfide Na₂S₂/Na₂S in liquid electrolyte. The high resistance of sulfur and its intermediate products (polysulfides) results in the unstable electrochemical contact in the sulfur electrode. In addition, during the charging process, the dissolved polysulfides shuttle between anode and cathode, including side reduction reaction with Na anode and reoxidation reaction with cathode. These problems lead to low utilization of active material, poor cycle life and low Coulombic efficiency. Obviously, it is impossible to meet all the requirements of the traditional Na-S battery. In order to form sulfur composite materials with good structure and properties, many strategies have been developed to improve discharge capacity, cycling and Coulombic efficiency. Other methods under study include new cell structures with trap intercalations, sodium/dissolved polysulfide cells, and effective electrolytes. Here, in this work we developed a sulfur host, which is hollow, polar and has effective morphology to accommodate sulfur and capture polysulfides by physiochemical adsorption. We have performed several tests to confirm the suppression of shuttle effect by BPCS i.e., visible adsorption method, UV-visible adsorption method and by XPS analysis of used separator as well as hydrophilicity of host material checked by water contact angle measurement, as shown in Figure 4.

In order to understand the source of polysulfide suppression, we conducted visible absorption test (Figure 4a) and ultraviolet visible light absorption test (Figure 4b). The results show that BPCS can adsorb Na₂S₆. In addition, we also perform XPS of the sulfur of the cycled separator to check the polysulfide dissolved in the electrolyte, as shown in Figure 4c (C3), the separator of BPCS@S cell has no polysulfide peak in the 165-160 eV region, while other composite separator have polysulfide peaks (C1 and C2 for BPCTE@S and BPSE@S, respectively). To check the hydrophilicity of BPCS, we measured the water contact angle, which is very low,

indicating the hydrophilicity of BPCS to adsorb polysulfides resulting in good electrochemical performance (**Figure 4d**).

Figure 4. (a) UV-visible absorbance spectra of Na₂S₆ before and after adding BPCS and AB, (b) Digital image of entrapment of Na₂S₆ by BPCS and AB, (c) XPS spectra of sulfur of separator after cycling and (d) water contact angle of BPCS.

Q4. Of course a big flaw are the limited experimental details. No electrolyte/sulfur ratio given. That's a step backwards in reporting accurate data. With a whatman separator (thick) an excess of electrolyte is guaranteed! The huge capacity loss in the first cycle confirms this view (dissolution). And in a flooded cell PS diffusion takes longer. And the glass fibers also adsorb the PS, and affect crystallization.

It would be good to test the material also with a PE separator and compare it to a Ketjen Black based reference cathode. And check performance under lean conditions with E/S ratios varying between 5 and 10.

A4. Thank you very much for the very professional comment. This question is closely related to Q3 from Reviewer 1. Please see our response to that question, as well as minor additional points here.

Figure 5. Charge-discharge profiles of KB@S/PE (a), BPCS@S/PE (b) and BPCS@S/Glass fiber (c).

Figure 5c shows the discharge plateaus of BPCS@S cathode are at higher potential than KB@S that indicates the suppression of electrochemical polarization. In addition, the plateaus of KB@S are sharper than BPCS@S which is due high electrical conductivity of KB (Figure 5a). As discussed previously, Figure 2 shows the clear performance comparison of glass fiber and Celgard polymer separator, as well as performance comparison of BPCS@S and KB@S composites.

To summarize, the glass fiber separator has high porosity and wettability, improves the absorption of dissolved polysulfide, slows down the diffusion process, and has good electrochemical performance.²

Furthermore, to check the performance of BPCS@S electrode, utilization of sulfur and effect of electrolyte volume used, we check the performance of BPCS@S at different E/S ratios, as shown in Figure 6. Figure 6 shows that when we apply an E/S ratio of 7:1 to 10:1, the battery capacity is acceptable. The capacity fade at low E/S ratio can be explained by Na-S reaction

mechanism. At low E/S ratio, the concentration and viscosity of polysulfides in electrolyte increase severely in the process of discharge, resulting in the increase of battery resistance and the discharge stops.

Figure 6. The effect of electrolyte/sulfur (E/S) ratio on discharge capacity of cathode (a) and discharge profiles of cathode at different E/S ratio using PE separator.

Q5. The in situ analysis is interesting. However, do the authors think these solid phase intermediates take part in the conversion mechanism, or are just side products? The catalytic effect is probably only important for dissolved species.

A5. The solid phase intermediates in the discharge process is Na₂S₂, which reduced to Na₂S in the discharge process.³ According to the theoretical analysis, the conversion of Na₂S₂ to Na₂S will contribute about half of the theoretical capacity. However, the dynamics of this process is poor, so electrocatalysts can be used to accelerate this process.⁴ The discharge process can be divided into four stages, which are the solid-liquid process, the liquid-liquid process, the liquid-solid process, and the solid-solid process.⁵ At present, various electrocatalysts have been adopted to promote these four-conversion processes.^{6, 7, 8, 9}

Therefore, electrocatalysis cannot only act on soluble polysulfides.

Q6. Technical problem in Adsorption data and porosity analysis. This material has no high

surface area, as one can see from the N₂ isotherms (Fig. S8) Fig. S8 probably shows a N₂ adsorption isotherm at 77 K, but is labeled “surface area“. Hence, the author is not familiar with this technique. Moreover, the isotherm does not close, probably not enough mass used for measurement. Repeat for (BPCS). Pore size analysis is meaningless.

A6. We appreciate the reviewer for a very instructive comment. We are sorry for ambiguity.

The already given data of adsorption and porosity is of CoTe₂ (BPTE) by mistake, it has low surface area because, it synthesized at high temperature (telluridation at 700 °C). Now we repeat and present the N₂ adsorption data for BPCS as shown in Figure 8 and Figure S13.

“...The Brunauer-Emmett-Teller (BET) surface area of BPCS is 158 m² g⁻¹ and the hysteresis loop is of type H4 indicates the physisorption isotherm of type I, which conforms to IUPAC, demonstrating that BPCS is composed of uniform slit-like pores, representing microporous structure. The microporous and ultramicroporous materials adsorb polysulfides more efficiently than other porous structures.¹⁰ ...”

Furthermore, the hysteresis loop is open at low relative pressure. This is due the microporous structure and swelling/shrinking of the material with the interaction of N₂. The relatively low N₂ desorption may be due to the too small micropores to desorb N₂.

Figure 7. N₂ physisorption isotherm (a) and pore size distribution (b) of BPCS.

Q7. The bipyramidal aspect is overemphasized. The crystals look almost like prism. Does the shape really play a role?

A7. Thank you for your comment. The size and shape of the host materials plays a role in the electrochemical performance.^{11, 12, 13, 14, 15, 16} In addition, during the conversion of S to Na₂S an expansion in volume occurs (i.e., decreases in volume during subsequent devulcanization). Therefore, in order to obtain a practical Na-S battery, it is necessary to design an appropriate electrode morphology to provide the necessary conductivity and to buffer the volume change. The hollow BPCS prisms have an internal space of 376 nm wide, which is sufficient to accommodate S and polysulfides in it to and to alleviate the volume expansion, as shown in systematic illustrated Figure 8.

Figure 8. Systematic volume expansion during lithiation.

Q8. Conversions are nicely depicted in the supporting. This scheme would be helpful for the reader in the main text.

A8. Thank you very much for your valuable suggestion. We have transferred the conversion scheme from supporting information to main text (Figure 1a).

Q9. XRD: sulfur, alpha or beta?

A9. We use α -S₈/orthorhombic sulfur. The XRD card of α -S₈/orthorhombic sulfur is PDF#08-0247, which is already shown in the sulfur composite XRD figures (Figure 3e). Long chain polysulfides are reduced to Na₂S₂ and Na₂S, and Na₂S₂ is further reduced to Na₂S. During the charging process, Na₂S is oxidized to soluble long chain polysulfides (Na₂S_x, 3 ≤ x ≤ 8), and then oxidized to β -S₈.¹⁷

Q10. What is TAA?

A10. TAA is the abbreviation of thioacetamide (source of sulfur). We have written the full name of the abbreviation in the main text.

Reviewer #3 (Comments to the Author): =====

Reviewer #3: I support the publication of the DFT sections, if the other reviewers endorse the experimental sections.

Response: Thank you very much for your affirmation of our work and positive comments for the manuscript. According to your suggestions, we have made the following explanations and modifications.

Specific comments:

Q1. On the “catalytic” description of the BPCS. I think a description of why the material is described as catalytic is warranted near the beginning of the paper, early in the introduction, because otherwise I am unsure why this description is so necessary. Don’t all cathodes “catalyze” reactions?

A1. Thank you for your professional comment. Room temperature Na-S batteries have great potential as a stationary energy storage battery in small and large power grids. However, due to the sluggish reaction kinetics, its energy and rate performance are limited, and its practical application is far from the theoretical value. To solve this problem, which is largely a problem of kinetics, a method based on catalysis is proposed. In this strategy, the catalytic sulfur host promotes the inter-conversion of polysulfides, speeds up the kinetics, alleviates the shuttle

effect, and demonstrates good electrochemical performance. Therefore, it is necessary to use a catalytic sulfur host material to allow the sulfur cell to reach a higher energy density and move towards a practical sodium sulfur cell. First, we must distinguish between "catalyst" and "electrocatalyst". The catalyst is added to the electrode material to promote the reaction, while the electrocatalyst itself is the electrode to promote the reaction. Specifically, in sulfur-batteries the electrocatalyst sulfur host materials give good electrochemical performance by adsorbing dissolved polysulfides and catalyze the inter-conversion reactions of polysulfides. For instance, carbon and other non-polar sulfur host materials (cathodes), such as carbon nanotubes, carbon nano fibers, carbon hollow nanospheres and double-shell carbon microspheres can act as sulfur host materials, but they don't have absorption ability and electrocatalysis capacity for polysulfides.^{18, 19, 20, 21} We can only prove that some materials can improve the performance, but we cannot be sure that all the host materials have catalytic properties for polysulfide conversion.

Q2. Why is cobalt used rather than another metal; is the choice of cobalt important? I don't think the choice of cobalt versus other transition metals is discussed in the paper and the computational authors could test other metal ions!

My general comments on the DFT section is that it's too long. For example, the section on the charge density differences (Fig 7b) are unsurprising and not that informative; the magnetic moment calculation may not be of interest to a general audience. Some of the DFT section should be moved to the supplemental information because the gas-phase simulations may not be such a good proxy for the real reactions in the electrolyte, 1M NaClO₄ in tetra-ethylene glycol dimethyl ether. For example, the charge density differences could look at lot different if explicit solvent was included.

A2. Cobalt active metal centers have higher catalytic activity than other metals.^{22, 23, 24} In

order to design this unique morphology, cobalt plays an important role, because its morphology and porosity are of great significance to sulfur battery.¹⁰

In the computational section of the paper, the analysis was restricted to an in-depth study of the CoX_2 family of dichalcogenide materials ($X=\text{S}, \text{Se}$ and Te) to provide a detailed analysis of the effect of anion chemistry on the catalytic performance of the material. The implicit solvation surface calculations used in this work, while computationally quite intensive, could be generalized to study the effect of other transition metal ions on the electrochemical performance of Na-S batteries, although this is beyond the scope of the current work. The following sentences have been added to the paper to highlight why Co was chosen in this study and how the computational approach could be useful for studying other transition metal systems.

- experimental and computational methodology used in this work is widely applicable to other transition metal dichalcogenide systems.^{22, 25}

We appreciate that there is currently an extensive discussion of the magnetic properties and charge distribution of the main text and as such, we have moved some of the discussion and Figure 7b and Figure 7b to the SI and added the following sentence to the main text

- As discussed further in Figure S25, the strong binding of the NaPSs to the surface is related to charge transfer between the undercoordinated Co^{2+} sites on the surface to the S atoms in the Na_2S_y cluster.

Q3. In the discussion section, line 438, the authors state, “DFT calculations support the mechanism that sulfide adsorption is superior for homogenous metal sulfides...when compared to the carbon host.” Unless I’m mistaken, this is one of the first times

“homogenous” is used to describe the surfaces and it’s unclear what the authors mean by “homogeneous”. Otherwise, the statement seems well backed up by their calculations.

A3. Thank you for your comment. We use the term "homogeneous" to refer to the chemical composition and the unified surface of the BPCS. We have now better explained the meaning of homogeneous in that sentence.

Q4. The authors do not comment on using a Hubbard-U term to correctly capture the electronic structure due to the cobalt ions. XPS of the cobalt has been performed (Figure 3), so a comment would be beneficial. Also, since the magnetic moment is described in detail, the effect of a Hubbard U should be discussed.

A4.

The reviewer makes a valuable point that an empirical Hubbard U correction (DFT+U) is often used to improve the description of the magnetic properties in transition metal oxide systems, however in the case of CoS₂, it has previously been shown (Kwon, S.K., et al. 2000. *Physical Review B*, 62(1), p.357.) that standard local spin density approximation (LSDA) calculations give better agreement with experimentally than LSDA+U calculations. The addition of a U parameter to selenide and telluride systems is also not routinely done in the literature. To consistently compare the results between the systems, we opted for the recently developed SCAN (+rvv10) functional that has been shown to give accurate results for transition metal systems, with results that only show a weak dependence of the Hubbard U parameter (Gautam, G.S. and Carter, E.A., 2018. *Phy. Rev. Mater.*, 2(9), p.095401.). The following sentences and references have been added to SI to highlight these points.

- The addition of an empirical Hubbard U correction (DFT+U) is commonly in the literature to improve the description of electron correlation in transition metal oxide systems, however it has previously been shown that standard local spin density

approximation (LSDA) calculations result in a better description of the magnetic properties in comparison with experiment than LSDA+U calculations (Kwon, S.K., et al. 2000. *Physical Review B*, 62(1), p.357.). Unlike in oxide systems, (Wang, L., Maxisch, T. and Ceder, G., 2006. *Phys. Rev. B*, 73(19), p.195107.) there is also limited data about calculation of empirical U values for selenide and telluride systems. The SCAN+rvv10 functional was therefore adopted in this work as it has been shown to give a good description of the electronic structure of highly correlated systems without a strong dependence on the Hubbard U parameter. (Gautam, G.S. and Carter, E.A., 2018. *Phy. Rev. Mater.*, 2(9), p.095401.)

Q5. The authors do a thorough search for the low energy surfaces of CoX₂ materials to create slab models. The authors do not clearly comment on why CoS₂ is chosen over CoSe₂ and CoTe₂ for making the BPCS. Is the sulfur in the cathode necessary for good performance? What is the take-away message from doing the selenium and tellurium calculations?

A5. In this study all three compounds CoS₂, CoSe₂ and CoTe₂ were fabricated as bipyramidal prisms, referred to as BPCS, BPCSE and BPCTE, respectively. As shown in SI Figure S19 and Figure 20, the most reversible electrochemical performance was achieved for the CoS₂ system. The calculations demonstrated that all of CoX₂ materials bind Na_xS_y polysulfides stronger than graphene/graphite, which explains the improved electrochemistry of these materials. The calculations also show that the structure/surface termination of the CoX₂ particles has more of an influence on the Na_xS_y binding than the nature of the anion, which is important as CoS₂ adopts the pyrite structure and CoSe₂ and CoTe₂ adopts the marcasite structure. We have added the following sentence to the text to clarify this:

- This result is in line with the experimental electrochemical results in Figure S20,

where the marcasite based BPCSE and BPCTE structures have similar rate performance, whereas the pyrite based BPCS structure has markedly better rate performance.

Q6. How do the authors ensure that they've left enough vacuum space to not have finite-size effect errors? Especially with the dipoles created on the surface due to the NaPSs molecules, 15 Å MAY NOT BE ENOUGH.

A6. The reviewer makes a valuable point that the vacuum thickness used in the calculations can affect the quality of the slab calculations used in this work. To check the vacuum thickness, we calculated the binding of the Na₂S₄ molecule to the CoS₂ slab with a vacuum thickness of 30 Å and found that the binding energy varied by less than 5 %. We therefore used a vacuum thickness of 15 Å for the remainder of the calculations as a balance between accuracy and computational cost. We have added the following sentence to the computational methods section in the SI to highlight this:

- Tests were performed with a larger vacuum thickness of ~30 Å for the binding of the Na₂S₄ molecule on the CoS₂ (100) surface and it was found that the binding energies varied by less than 5 %. A vacuum thickness of ~15 Å was therefore chosen for all subsequent calculations as a balance between accuracy and computational efficiency.

Q7. Line 361: It is interesting that the author choose to calculate the formation of free Na₂S₇ molecules in the presence of an implicit solvent using VASPsol. They should include an implicit solvent for the formation energies on graphene and p_CoS₂ as well, or explain why they did not do these calculations. Figure 7 would be much more convincing and clear if implicit solvent were included as a comparison. Currently, the point of Figure 7 is confusing. There is a positive formation energy in the liquid and on graphene – indicating that the

compounds should not form. Some of the formation energies on p-CoS₂ are quite small – would they be positive if solvent was included? Since trends are reported, it is acceptable, but not the best possible results, that Hubbard-U and implicit solvent are not considered when calculating the formation energy for formation on CoS₂.

A7. The reviewer makes an important point that implicit solvation will affect the binding energies on the different surfaces. For all of the surface structures in Figure 7a (CoX₂ and graphene), implicit solvation was included to allow for a direct comparison of the results. We have added an additional sentence to the methods sections to make this clearer:

- The effects of solvation on the slabs with and without the bound NaPSs were taken into account via the inclusion of an implicit solvation model in VASPsol.

Q8. Line 381: I do not see where the following result is shown in a plot or the value of binding energy explained, “The binding energy of NaPSs on the p-CO₂ (100) surface increases systematically as the chain length decreases, reaching 2eV for the Na₂S molecule.” It is certainly not clear from Fig. 7 a).

A8. Reviewer 3 makes a good point that the ‘binding energy’ was not clearly defined in the text. The sentence has therefore been changed to:

- The difference in the energy between the NaPS bound to the p-CoO₂ (100) surface and the ‘liquid’ state increases systematically as the chain length decreases, reaching 2 eV for the Na₂S molecule.

References

1. Luo X, Pan W, Liu H, Gong J, Wu H. Glass fiber fabric mat as the separator for lithium-ion battery with high safety performance. *Ionics* **21**, 3135-3139 (2015).
2. Zhu J, et al. Understanding glass fiber membrane used as a novel separator for lithium–sulfur batteries. *J. Mem. Sci.* **504**, 89-96 (2016).
3. Li Y, et al. Fast conversion and controlled deposition of lithium (poly)sulfides in

- lithium-sulfur batteries using high-loading cobalt single atoms. *Energy Storage Mater.* **30**, 250-259 (2020).
4. Yang X, et al. Promoting the Transformation of Li₂S₂ to Li₂S: Significantly Increasing Utilization of Active Materials for High-Sulfur-Loading Li–S Batteries. *Adv. Mater.* **31**, 1901220 (2019).
 5. Song Y, Cai W, Kong L, Cai J, Zhang Q, Sun J. Rationalizing Electrocatalysis of Li–S Chemistry by Mediator Design: Progress and Prospects. *Adv. Energy Mater.* **10**, 1901075 (2020).
 6. Fan FY, Carter WC, Chiang Y-M. Mechanism and Kinetics of Li₂S Precipitation in Lithium–Sulfur Batteries. *Adv. Mater.* **27**, 5203-5209 (2015).
 7. Kong L, et al. Current-density dependence of Li₂S/Li₂S₂ growth in lithium–sulfur batteries. *Energy & Environ. Sci.* **12**, 2976-2982 (2019).
 8. Lin H, et al. Electrocatalysis of polysulfide conversion by sulfur-deficient MoS₂ nanoflakes for lithium–sulfur batteries. *Energy & Environ. Sci.* **10**, 1476-1486 (2017).
 9. Yuan H, et al. Conductive and Catalytic Triple-Phase Interfaces Enabling Uniform Nucleation in High-Rate Lithium–Sulfur Batteries. *Adv. Energy Mater.* **9**, 1802768 (2019).
 10. Hippauf F, et al. The Importance of Pore Size and Surface Polarity for Polysulfide Adsorption in Lithium Sulfur Batteries. *Adv. Mater. Interfaces* **3**, 1600508 (2016).
 11. Zhang L, Chen L, Zhou X, Liu Z. Morphology-Dependent Electrochemical Performance of Zinc Hexacyanoferrate Cathode for Zinc-Ion Battery. *Sci. Rep.* **5**, 18263 (2015).
 12. Gan Q, He H, Zhao K, He Z, Liu S. Morphology-dependent electrochemical performance of Ni-1,3,5-benzenetricarboxylate metal-organic frameworks as an anode material for Li-ion batteries. *J. Colloid Interface Sci.* **530**, 127-136 (2018).
 13. Kong L, et al. Aggregation-Morphology-Dependent Electrochemical Performance of Co₃O₄ Anode Materials for Lithium-Ion Batteries. *Molecules* **24**, 3149 (2019).
 14. Li Z, Ding S, Yin J, Zhang M, Sun C, Meng A. Morphology-dependent electrochemical performance of VS₄ for rechargeable magnesium battery and its magnesiation/demagnesiation mechanism. *J. Power Sources* **451**, 227815 (2020).
 15. Uke SJ, Chaudhari GN, Bodade AB, Mardikar SP. Morphology dependant electrochemical performance of hydrothermally synthesized NiCo₂O₄ nanomorphs. *Mater. Sci. Energy Technol.* **3**, 289-298 (2020).
 16. Xie F, Zhou M, Wang G, Wang Q, Yan M, Bi H. Morphology-dependent electrochemical performance of nitrogen-doped carbon dots@polyaniline hybrids for supercapacitors. *Inter. J. Energy Research* **43**, 7529-7540 (2019).

17. Deng C, Wang Z, Wang S, Yu J. Inhibition of polysulfide diffusion in lithium–sulfur batteries: mechanism and improvement strategies. *J. Mater. Chem. A* **7**, 12381-12413 (2019).
18. Hwang TH, Jung DS, Kim J-S, Kim BG, Choi JW. One-Dimensional Carbon–Sulfur Composite Fibers for Na–S Rechargeable Batteries Operating at Room Temperature. *Nano Lett.* **13**, 4532-4538 (2013).
19. Lu Q, et al. Freestanding carbon fiber cloth/sulfur composites for flexible room-temperature sodium-sulfur batteries. *Energy Storage Mater.* **8**, 77-84 (2017).
20. Wang Y-X, et al. Achieving High-Performance Room-Temperature Sodium–Sulfur Batteries With S@Interconnected Mesoporous Carbon Hollow Nanospheres. *J. Am. Chem. Soc.* **138**, 16576-16579 (2016).
21. Zhang L, et al. Self-Assembling Hollow Carbon Nanobeads into Double-Shell Microspheres as a Hierarchical Sulfur Host for Sustainable Room-Temperature Sodium–Sulfur Batteries. *ACS Appl. Mater. Interfaces* **10**, 20422-20428 (2018).
22. Yuan Z, et al. Powering Lithium–Sulfur Battery Performance by Propelling Polysulfide Redox at Sulfiphilic Hosts. *Nano Lett.* **16**, 519-527 (2016).
23. Choi I, et al. Facile Synthesis of M-MOF-74 (M=Co, Ni, Zn) and its Application as an ElectroCatalyst for Electrochemical CO₂ Conversion and H₂ Production. *J. Electrochem. Sci. Technol.* **8**, 61-68 (2017).
24. Lee H, Hong JA. Enhancement of Catalytic Activity of Reduced Graphene Oxide Via Transition Metal Doping Strategy. *Nanoscale Research Lett.* **12**, 426 (2017).
25. Peng H-J, et al. Enhanced Electrochemical Kinetics on Conductive Polar Mediators for Lithium–Sulfur Batteries. *Angew. Chem. Int. Ed.* **55**, 12990-12995 (2016).

REVIEWERS' COMMENTS:

Reviewer #1 (Remarks to the Author):

Ok to publish.

Reviewer #2 (Remarks to the Author):

The authors have addressed most of the referee comments well. Regarding the Whatman separator and E/S ratio, it turns out that only high E/S ratios work and the Whatman separator (very thick) is mandatory. For practical applications this is a severe limitation showing that the role of the porous materials is overemphasized (flooded cell). The relevance and impact of this work may thus be limited.

The N₂ adsorption hysteresis of BPCS is still difficult to understand. Instead of swelling it looks more like a kinetic effect due to the hindered diffusion through the shell during ad- and desorption. In any case it is quite vague to derive a pore size distribution (neither for a swelling system nor for a kinetically hindered system PSD can apply).

Reviewer #3 (Remarks to the Author):

My review was for the DFT section of the paper. The authors have addressed my concerns and fixed confusing parts of the computational section. I only have a few more suggestions, but publication is not contingent on the authors taking my suggestions.

If the editors agree that the experimental reviewers concerns were satisfied, I support publication.

Q1: The authors explained the use of catalytic in their response. I suggest they explicitly note in the main text that the cathodes (18-21) are non-catalytic, such as the sentence at the bottom of page 4.

Q2: The authors have tidied up the DFT section by moving parts to the SI and also included a comment that cobalt was chosen over other transition metals, perhaps inspiring future work.

Q3: The authors have answered by question about homogenous. Perhaps they should write, "interwoven surfaces and chemical composition" to indicate that the chemical composition is also homogenous.

Q

4: I am satisfied with the explanation of the SCAN+rvv10 functional in the SI section.

Q5: The additional sentences helps explain the calculations.

Q6: I am satisfied with the addition to the SI comparing two vacuum sizes.

Q7: The additional sentence helps clarify how the implicit solvent was used in the calculations.

Q8: I now understand better what is the point of Figure 7!

I hope that the BPCS catalysts really does advance the development of Na-S battery technologies!

Reviewers' comments and our Response:

Reviewer #1 (Comments to the Author): =====

Comment: Ok to publish.

Response. Thank you very much for your affirmation of our work and your favorable comments on the manuscript.

Reviewer #2 (Comments to the Author): =====

Comment. The authors have addressed most of the referee comments well. Regarding the Whatman separator and E/S ratio, it turns out that only high E/S ratios work and the Whatman separator (very thick) is mandatory. For practical applications, this is a severe limitation showing that the role of the porous materials is overemphasized (flooded cell). The relevance and impact of this work may thus be limited.

Response. We are very happy to see that we have addressed most of the points in the previous revision to your satisfaction and we appreciate your further inputs.

At present, some valuable conclusions have been obtained in the study of the E/S ratio. Most studies use high E/S ratios and discuss the limitations of low E/S ratios. When the electrolyte content is low, the polysulfides concentration in the electrolyte rises to a critical value. The interaction between the corrosive electrolyte (because of high concentration of polysulfides) and the metal anode causes the anode to corrode, resulting in the loss of active material (sulfur) on the anode surface.¹ A low E/S ratio decreases the utilization of sulfur. It is worth noting that there have been some previous examples to demonstrate that the E/S ratio must be ≥ 10 for sulfur batteries in order to have a good power capability.

- I. Emerce et al.² studied that the specific energy increased significantly to 9-10 $\mu\text{L}/\text{mg}$ E/S ratio.
- II. Hagen and co-workers also mentioned that capacity is acceptable for E/S above 7 $\mu\text{L}/\text{mg}$.
- III. Sun et al.³ examined that, to build sulfur cell with good power capability the E/S ratio must be higher than 10 $\mu\text{L}/\text{mg}$.
- IV. Zhang et al.⁴ have investigated that the appropriate E/S ratio for sulfur cell is 10 $\mu\text{L}/\text{mg}$.
- V. Zheng and co-workers¹ also have investigated that the optimized E/S ratio for sulfur cell is 20 $\mu\text{L}/\text{mg}$.
- VI. Choi's research group⁵ also have indicated that 10 $\mu\text{L}/\text{mg}$ E/S ratio is suitable for sulfur battery.

According to the E/S ratio of lithium sulfur battery in the above-mentioned literature and according to the Huang et al.⁶ most of the recently published literatures on Li-S batteries use E/S greater than 10.

In addition, Wang Guoxiu et al.⁷ published an article on Na-S battery in Nature Communications, and they used glass fiber separator as well as high electrolyte volume (20 $\mu\text{L}/\text{mg}$ E/S ratio). Our results are superior to those reported previously, indicating that the Na-S battery has superior performance in the range of 7-10 $\mu\text{L}/\text{mg}$ E/S.

Comment. The N_2 adsorption hysteresis of BPCS is still difficult to understand. Instead of swelling, it looks more like a kinetic effect due to the hindered diffusion through the shell during ad- and desorption. In any case, it is quite vague to derive a pore size distribution (neither for a swelling system nor for a kinetically hindered system PSD can apply).

Response. Thank you so much for your suggestion and we appreciate your further input. We

try to solve the problem of hysteresis loop by optimizing degassing time and degassing temperature, and use another analyzer with good performance.

Last time, we performed N₂ adsorption and porosity measurements on the Quantachrome QUADRASORB automatic surface area and pore size analyzer. The sample was degassed at room temperature for 12 hours. However, this time we measured on another machine Micromeritics ASAP 2020 HD88 (Figure 2) with the sample degassed at 120 °C for 24 hours, which provided favorable results for the hysteresis loop, as shown in the following Figure 1 and supplementary Figure 13.

“...The Brunauer-Emmett-Teller (BET) surface area of BPCS is 119.9 m² g⁻¹ and the hysteresis loop is of type H4 and indicates the physisorption isotherm of type I, which conforms to IUPAC, demonstrating that BPCS is composed of uniform slit-like pores, representing microporous patterns in structure. The microporous materials adsorb polysulfides more efficiently than other porous structures.⁸ ... (See Page 6 Line 18)”

Figure 1. N₂ physisorption isotherm (a) and pore size distribution (b) of BPCS.

Figure 2. Digital photograph of automatic surface area and pore size analyzers.

Reviewer #3 (Comments to the Author): =====

Comment: The authors have addressed my concerns and fixed confusing parts of the computational section. I only have a few more suggestions, but publication is not contingent on the authors taking my suggestions. If the editors agree that the experimental reviewers concerns were satisfied, I support publication.

Response: Thank you very much for your affirmation of our work and positive comments for the manuscript. We are very happy to see we have addressed most of the points in the previous revision to your satisfaction and we appreciate your further inputs.

Comment. The authors explained the use of catalytic in their response. I suggest they explicitly note in the main text that the cathodes (18-21) are non-catalytic, such as the sentence at the bottom of page 4.

Response. Thank you very much for your valuable suggestion and we appreciate your further input. I think we should be careful when saying that other materials are ‘non-catalytic’ because if it binds the NaPSs, it is likely to have some effect on the kinetics of NaPSs conversion. We have already discussed this point in the previous response letter. However, we have now clarified this point in the main text as follow:

“...As a result, hollow polar S hosts can more efficiently block NaPSs diffusion than other structures, such as nano-particles and flakes. In addition to the above strategies, the use of host materials that effectively catalyse the conversion of long-chain NaPSs (Na_2S_x , $4 \leq x \leq 8$) to short chain NaPSs, which is a particularly promising approach to inhibit NaPSs diffusion.⁹ Due to the insulating properties of sulfur and NaPSs, the electrochemical discharge/charge processes are sluggish. For non-catalytic hosts such as carbon, carbon nanotubes, carbon nano-fibers, carbon hollow nanospheres and double-shell carbon microspheres, the conversion of NaPSs is slow and the intermediate polysulfides can easily dissolve into the electrolyte. However, due to the use of the catalytic S host, such as electronically conducting transition metal sulfide hosts can effectively act as electrocatalysts to accelerate the redox kinetic of long chain NaPSs (Na_2S_x , $4 \leq x \leq 8$) and efficiently convert to solid phase S and $\text{Na}_2\text{S}/\text{Na}_2\text{S}_2$... (See Page 4 Line 18)”

Comment. The authors have tidied up the DFT section by moving parts to the SI and also included a comment that cobalt was chosen over other transition metals, perhaps inspiring future work.

Response. Thank you very much for your affirmation of our reply. We are glad to see that we have addressed your point.

Comment. The authors have answered by question about homogenous. Perhaps they should

write, “interwoven surfaces and chemical composition” to indicate that the chemical composition is also homogenous.

Response. We are glad to see that we have addressed your point. We have modified it as “interwoven surfaces and chemical composition”... (See Page 16 Line 13).

Comment. I am satisfied with the explanation of the SCAN+rvv10 functional in the SI section.

Response. Thank you very much for your affirmation of our reply.

Comment. The additional sentences help explain the calculations.

Response. Thank you very much for your affirmation of our reply.

Comment. I am satisfied with the addition to the SI comparing two vacuum sizes.

Response. Thank you very much for your affirmation of our reply.

Comment. The additional sentence helps clarify how the implicit solvent was used in the calculations.

Response. Thank you very much for your affirmation of our reply.

Comment. I now understand better what is the point of Figure 7!

Response. Thank you very much for your affirmation of our reply. We are glad to see that we have addressed your point.

Comment. I hope that the BPCS catalysts really does advance the development of Na-S battery technologies!

Response. Thank you very much for your affirmation of our work.

References

1. Zheng J, *et al.* How to obtain reproducible results for lithium sulfur batteries? *J Electrochem Soc* **160**, A2288 (2013).
2. Emerce NB, Eroglu D. Effect of Electrolyte-to-Sulfur Ratio in the Cell on the Li-S Battery Performance. *J Electrochem Soc* **166**, A1490-A1500 (2019).
3. Sun K, *et al.* Effect of Electrolyte on High Sulfur Loading Li-S Batteries. *J Electrochem Soc* **165**, A416-A423 (2018).
4. Zhang SS. Improved cyclability of liquid electrolyte lithium/sulfur batteries by optimizing electrolyte/sulfur ratio. *Energies* **5**, 5190-5197 (2012).
5. Choi J-W, Kim J-K, Cheruvally G, Ahn J-H, Ahn H-J, Kim K-W. Rechargeable lithium/sulfur battery with suitable mixed liquid electrolytes. *Electrochim Acta* **52**, 2075-2082 (2007).
6. Hagen M, Fanz P, Tübke J. Cell energy density and electrolyte/sulfur ratio in Li-S cells. *J Power Sources* **264**, 30-34 (2014).
7. Xu X, *et al.* A room-temperature sodium-sulfur battery with high capacity and stable cycling performance. *Nat Commun* **9**, 3870 (2018).
8. Hippauf F, *et al.* The Importance of Pore Size and Surface Polarity for Polysulfide Adsorption in Lithium Sulfur Batteries. *Advanced Materials Interfaces* **3**, 1600508 (2016).
9. Li L, *et al.* Phosphorene as a Polysulfide Immobilizer and Catalyst in High-Performance Lithium-Sulfur Batteries. *Adv Mater* **29**, 1602734 (2017).